# Proteomic Characterization of the *Clostridium cellulovorans* Cellulosome and Noncellulosomal Enzymes with Sorghum Bagasse

**DOI:** 10.3390/ijms262311728

**Published:** 2025-12-03

**Authors:** Mohamed Y. Eljonid, Fumiyoshi Okazaki, Eiji Hishinuma, Naomi Matsukawa, Sahar Hamido, Yutaka Tamaru

**Affiliations:** 1Department of Life Sciences, Graduate School of Bioresources, Mie University, 1577 Kurimamachiya-cho, Tsu 514-8507, Japan; 522d3s2@m.mie-u.ac.jp (M.Y.E.); okazaki@bio.mie-u.ac.jp (F.O.); 2Advanced Research Center for Innovations in Next-Generation Medicine, Tohoku University, 2-1 Seiryo-machi, Aoba-ku, Sendai 980-8573, Japan; eiji.hishinuma.e7@tohoku.ac.jp; 3Tohoku Medical Megabank Organization, Tohoku University, 2-1 Seiryo-machi, Aoba-ku, Sendai 980-8573, Japan; naomi.matsukawa.e5@tohoku.ac.jp; 4Green X-Tech Research Center, Green Goals Initiative, Tohoku University, 6-6-07 Aoba, Aramaki-aza, Aoba-ku, Sendai 980-8579, Japan; sahar.hamido.c4@tohoku.ac.jp; 5Department of Molecular Bioengineering, Graduate School of Engineering, Tohoku University, 6-6-07 Aoba, Aramaki-aza, Aoba-ku, Sendai 980-8579, Japan

**Keywords:** *Clostridium cellulovorans*, cellulosome, noncellulosomal proteins, proteomic analysis, LC-MS/MS

## Abstract

Sorghum, the fifth major global cereal, has potential as a source crop in temperate regions. To completely use sorghum bagasse, the ideal enzyme cocktail aims to identify and select the contributed enzymatic system. This study investigated the enzymatic system of *Clostridium cellulovorans* cellulosome and noncellulosomal enzymes using sodium dodecyl sulfate polyacrylamide gel electrophoresis (SDS-PAGE) and liquid chromatography–tandem mass spectrometry LC-MS/MS. Enzyme solutions from treated and untreated sorghum bagasse were prepared and compared based on carboxymethyl cellulase (CMCase) activity. As a result, the enzyme solution derived from untreated sorghum bagasse had the highest activity. Protein bands from each *C. cellulovorans* culture showed distinct patterns on SDS-PAGE examination: three enzyme fractions, including culture supernatants, crystalline cellulose (Avicel) bound, and unbound fractions. These results suggested that untreated sorghum bagasse induced a variety of cellulosomal and uncellulosomal proteins. On the other hand, 5% or 10% sorghum supernatants could not induce Avicel-bound proteins, including the cellulosome, although even 5% sorghum juice induced three major bands: 180 kilodalton (kDa), 100 kDa, and 70 kDa, respectively. In contrast, cellobiose induced three major bands, while the total number of all isolated proteins from the cellobiose medium was the most limited among all culture media. More intriguingly, our investigation detected one cellulosomal protein, hydrophobic protein A (HbpA) and three noncellulosomal enzymes, indicating that glycosyl hydrolase family 130 (GH130) was identified as a biomass-induced enzyme in good accord with previously published proteomic studies. Therefore, the proteomic dataset generated in this study provides us a foundation for future computational approaches, including machine learning-based prediction of optimal enzyme cocktails for target biomass degradation.

## 1. Introduction

The world’s expanding population necessitates an immediate solution to lower carbon dioxide emissions and increase the production of green chemicals. The conversion of non-edible and unused biomass into valuable bio-based products stands as a solid solution. However, its complex structure and the inefficiencies of traditional processing methods present challenges in its sustainable conversion. Cellulosic and herbaceous types of biomass (soft biomass), such as rice straw, switchgrass, and bagasse, show promise as substrates to produce chemical products and fuels [1]. Sorghum (*Sorghum bicolor* L.) is a vital source of nutrients in the human diet and ranks as the fifth most important food crop worldwide, with significant implications for human health [2]. In addition, sorghum has attracted strong interest because of its many good characteristics such as rapid growth and high sugar accumulation, high biomass production potential, excellent nitrogen usage efficiency, wide adaptability, drought resistance, waterlodging tolerance, and salinity resistance [3]. Sorghum varieties include grain sorghum, sweet sorghum, and biomass sorghum, and grain sorghum, having starch content equivalent to corn, has been considered as a feedstock for ethanol production. On the other hand, its sorgoleone content interestingly causes the inhibition of bacterial enzymes in soil that turn ammonium into nitrate, slowing nitrogen loss and promoting sustainable farming [4]. Furthermore, sweet sorghum juice contains sucrose, glucose, and fructose, which are readily fermented by Saccharomyces cerevisiae and hence are good substrates for ethanol fermentation.

The cellulosome is a multi-enzyme complex in which catalytic domains containing dockerin sequences (DS) bind to cohesin domains on the scaffolding protein, and carbohydrate-binding modules (CBM) facilitate substrate adhesion [5,6]. The approaches for the investigation of enzymatic effects were observed in *C. cellulovorans*. There have so far been reported on proteomic approaches to elucidate the enzymatic and metabolic systems in *C. cellulovorans* [1,7,8,9]. Morisaka et al. reported proteome analysis of the *C. cellulovorans* cellulosome after culture in 0.3% (*w*/*v*) cellobiose, 0.3% (*w*/*v*) Avicel, or 0.3% (*w*/*v*) xylan demonstrated the production of carbon source-adapted cellulosome components [7]. With the monolithic column, 679 non-redundant peptides were identified from 193 proteins, while the conventional column yielded 46 peptides from 26 proteins. In the identification of the scaffold protein CbpA, 26% of the sequence coverage, including some peptides that could not be detected using the conventional column, by which only 2% of the sequence coverage was identified, was performed by the monolithic column. On the other hand, based on the exoproteome analysis of *C. cellulovorans*, the protein Clocel_3197 belonging to glycosyl hydrolase (GH) family 130 was found to be commonly involved in the degradation of every natural soft biomass such as bagasse, corn germ, and rice straw [1]. Moreover, a total of 1895 cellular proteins from glucose, cellulose, xylan, galactomannan, and pectin media of *C. cellulovorans* were identified, among which 865 were common to all carbon sources [8]. Overall, 879 secreted proteins were identified, of which 361 were common to all carbon sources. Thus, the proteome analysis covered approximately 50% of all gene products of *C. cellulovorans*; this proteome coverage had so far been the highest one reported in *C. cellulovorans* studies. Finally, Usai et al. had reported focusing on soluble whole-cell extracts to identify proteins specifically associated with glucose or crystalline cellulose (Avicel) metabolism [9]. In this study, 621 proteins were quantified, corresponding to about 15% of the *C. cellulovorans* annotated proteins. To analyze the distribution of their biological functions, the quantified proteins were annotated by means of the Cluster of Orthologous Genes (COGs) categories (http://eggnogdb.embl.de/#/app/home, accessed on 28 November 2025). The large majority (522 proteins, 84%) of quantified proteins were associated with at least a known function grasped by a COG category, whereas the remaining 16% consisted of proteins with unknown functions. Of the total quantified proteins, 319 were found to be differentially expressed when comparing the two growth conditions; specifically, they were at least 1.5-fold more abundant (*p*-value < 0.05) in one growth condition compared to the other.

In our more recent study, we investigated the cellulolytic system of *C. cellulovorans,* mainly consisting of the cellulosome that synergistically collaborates with noncellulosomal enzymes by using cellulosic biomass such as shredded paper, rice straw, and sugarcane bagasse [10]. Regarding the rice straw and sugarcane bagasse, while the degradative activity of rice straw was most active using the cellulosome in the culture supernatant of rice straw medium, that of sugarcane bagasse was most active using the cellulosome from the supernatant of cellobiose medium. Furthermore, since we attempted to choose reaction conditions more efficiently for the degradation of sugarcane bagasse, a wet jet milling device together with L-cysteine as a reducing agent was used in the study.

In the field of energy section using terrestrial plants of Fukushima Institute for Research and Innovation (F-REI), we examine the production of green chemicals using sorghum as a biomass resource. Sorghum bagasse is focused on as the source of carbon-neutral materials to produce biobutanol with the consolidated bioprocessing (CBP) of *Clostridium* cocultivation. In this study, the chemical pretreatments of sorghum bagasse were carried out to yield soluble and fermentable sugars and to elucidate the extracellular proteins and enzymes from *C. cellulovorans* cultivated with several carbohydrate substrates as a carbon source. Furthermore, proteomic analysis of the protein bands from them by SDS-PAGE was carried out and compared with previous proteomic studies on *C. cellulovorans*.

## 2. Results

### 2.1. Evaluation of Chemical Pretreatment of Sorghum Bagasse Based on CMCase Activity

Batch cultivations in 100 mL serum bottles were performed containing 100 mL *C. cellulovorans* (C.c) medium supplemented with one of the five carbon substrates: 0.5% cellobiose, 1% filter paper, 1% untreated sorghum bagasse, 1% sulfuric acid–butanol treated sorghum bagasse, or 1% alkaline NaOH, which is sodium hydroxide (NaOH) treated sorghum bagasse. All cultures were maintained under anaerobic conditions at 37 °C without shaking. An inoculum of *C. cellulovorans* medium containing 0.5% cellobiose was harvested at 48 h post-inoculation, corresponding to the exponential growth phase, to initiate batch cultivations. These cultivations were harvested at 170 h for enzyme extraction, SDS-PAGE, and proteomic analyses aimed at capturing active enzymatic profiles during the stationary phase (28–29 h). Additionally, samples were collected at 15 time points throughout the 170 h fermentation period (0, 1, 16, 24, 32, 40, 48, 56, 64, 72, 84, 99, 123, 146, and 170 h) to assess temporal changes in enzymatic activity and sugar concentrations. Measurement of CMCase activity over the cultivation period revealed pronounced substrate-dependent variations in cellulolytic enzyme production (Table 1). Specific activity values, expressed as units per mg of total protein (U/mg), provided insights into the enzymatic quality and efficiency of protein production under different substrate conditions. Especially, untreated sorghum culture demonstrated the highest mean specific activity at 6.63 ± 0.82 U/mg protein (mean ± standard deviation calculated from 15 time points), representing optimal enzymatic quality among all tested conditions. The elevated specific activity suggested that the complex, unmodified lignocellulosic structure of untreated sorghum bagasse induced expression of highly active cellulolytic enzymes. On the other hand, filter paper, serving as a pure cellulose control, yielded a specific activity of 5.70 ± 0.71 U/mg, demonstrating efficient enzyme production on crystalline cellulose substrates. Cellobiose culture, despite rapid growth and high protein production, showed lower specific activity (4.70 ± 0.58 U/mg), indicating that soluble substrates may not optimally induce cellulolytic enzyme expression. The specific activity hierarchy (untreated sorghum bagasse > filter paper > cellobiose) indicated that substrate complexity positively correlates with enzymatic quality (Figure 1). Furthermore, pretreated sorghum substrates exhibited significantly reduced specific activities compared to untreated controls. Acid–butanol-treated sorghum bagasse showed specific activity of 4.11 ± 0.49 U/mg, representing a 38.0% reduction compared to untreated sorghum bagasse. Thus, the substantial decrease occurred despite comparable total protein production (370.6 μg/mL versus 340.6 μg/mL for untreated sorghum bagasse), indicating that acid–butanol-treated sorghum bagasse specifically affected enzymatic quality rather than quantity. On the other hand, alkaline-treated sorghum bagasse demonstrated the lowest specific activity at 3.59 ± 0.43 U/mg, a 45.8% reduction from untreated conditions. As a result, the progressive decrease in specific CMCase activity with pretreatment severity (untreated sorghum bagasse > acid–butanol-treated bagasse > alkaline-treated sorghum bagasse) indicated dose-dependent effects on extracellular enzymatic machinery.

Statistical analysis was performed to evaluate the significance of differences in (CMCase) activities between untreated sorghum bagasse and other carbon sources using independent samples *t*-tests, except for the filter paper-specific activity comparison, which required the Wilcoxon signed-rank test due to non-normal data distribution. Error bars representing the standard error of the mean (n = 15 time points) and statistical significance indicators are displayed in Figure 1. For volumetric activity, untreated sorghum bagasse demonstrated significantly higher activity compared to all other tested substrates. The comparison with cellobiose yielded *p* = 0.0497, indicating that, despite cellobiose being a readily fermentable disaccharide, untreated sorghum bagasse induced greater total enzyme secretion. Filter paper, representing pure crystalline cellulose, showed significantly lower volumetric activity than untreated sorghum bagasse (*p* = 0.0042), suggesting that the heterogeneous lignocellulosic composition of sorghum bagasse provides superior induction signals for cellulolytic enzyme production. The chemically pretreated sorghum substrates exhibited even more pronounced reductions in volumetric activity, with acid–butanol-treated sorghum bagasse showing *p* = 0.0034 and alkaline-treated sorghum bagasse demonstrating the most significant reduction (*p* < 0.0001). The extremely low *p*-value for alkaline-treated substrate indicates a fundamental impairment of the enzyme induction system caused by alkaline pretreatment. The specific activity analysis revealed distinct patterns that provide insight into enzymatic quality across substrates. Notably, the comparison between untreated sorghum bagasse and cellobiose showed no significant difference in specific activity (*p* = 0.1780). This observation can be explained by the presence of residual simple sugars in the sorghum bagasse. It has been well documented that mechanical juice extraction from sweet sorghum stalks does not collect all available sugars, and a considerable amount of residual soluble sugars (sucrose, glucose, and fructose) remains in the bagasse after squeezing [11,12]. Studies have demonstrated that up to 0.4 g of soluble sugars per gram of bagasse can be recovered by water extraction after mechanical pressing, and sweet sorghum bagasse typically contains approximately 25% water-extractable components, including residual fermentable sugars [12,13]. Therefore, the similar specific activity between untreated sorghum bagasse and cellobiose likely reflects the induction of comparable enzymatic profiles by the residual simple sugars present in the bagasse, which would trigger similar metabolic responses as the soluble disaccharide cellobiose. In contrast, filter paper showed significantly lower specific activity (*p* = 0.0353), suggesting that pure crystalline cellulose substrates without soluble sugar components induce enzymes with reduced catalytic efficiency compared to substrates containing fermentable sugars. Chemical pretreatments significantly reduced specific activity, with acid–butanol-treated sorghum bagasse demonstrating *p* = 0.0312 and alkaline-treated sorghum bagasse showing *p* = 0.0066. These pretreatments are known to remove not only lignin and hemicellulose but also residual soluble sugars, thereby eliminating the induction signals provided by these simple carbohydrates. The progressive decrease in specific activity with pretreatment severity (untreated > acid–butanol-treated > alkaline-treated) correlates with the increasing removal of both structural and soluble carbohydrate components, fundamentally altering the substrate composition sensed by *C. cellulovorans.* The temporal consistency of these statistical differences across all 15 measurement points from 0 to 170 h confirms that the observed activity patterns reflect fundamental substrate-enzyme relationships rather than transient phenomena. Collectively, these statistical analyses establish that untreated sorghum bagasse, with its complex composition, including residual simple sugars, cellulose, hemicellulose, and lignin, provides the optimal substrate for CMCase induction in *C. cellulovorans*.

### 2.2. Proteomic Analysis of Carbohydrate-Related Proteins and Their Comparison Based on Sorghum Bagasse with or Without Chemical Pretreatments

Anaerobic batch cultivations of *C. cellulovorans* were carried out in a 100 mL medium containing 0.5% cellobiose, 1% filter paper, 1% untreated sorghum bagasse, and 1% treated sorghum bagasse with acid–butanol or alkaline at 37 °C without shaking. SDS-PAGE analysis showed 120 kDa bands (upper nos. 3.1, 4.1, and 5.1) in 0.5% cellobiose and 1% untreated sorghum bagasse, respectively (Figure 2, lanes 1 and 3), while no or a slight 120 kDa band appeared in 1% filter paper (Figure 2, lane 2) and 1% acid–butanol treated sorghum bagasse and 1% alkaline (NaOH) treated sorghum bagasse (Figure 2, lanes 4 and 5). The selection of these specific molecular weight bands (120 kDa, 80 kDa, and 60 kDa) for LC-MS/MS analysis was strategically based on their differential expression patterns between cellulosic and soluble sugar substrates. These bands were prominently expressed in cultures containing filter paper and sorghum bagasse but were absent or barely detectable in the cellobiose medium, indicating biomass-specific induction. This targeted approach enabled efficient identification of cellulosomal and noncellulosomal enzymes specifically involved in lignocellulosic degradation, while excluding constitutively expressed housekeeping proteins that would not contribute to understanding sorghum bagasse-specific enzymatic machinery. In addition to 120 kDa bands, 80 kDa bands (lower nos. 3.1, 4.1, and 5.1) obviously appeared in 1% filter paper and 1% untreated or treated sorghum bagasse (Figure 2, lanes 2–5), while a slight 80 kDa band in 0.5% cellobiose was detected in SDS-PAGE (Figure 2, lanes 1). Furthermore, whereas 60 kDa bands (nos. 3.3, 4.3, and 5.3) were obviously found in 1% filter paper and 1% untreated or treated sorghum bagasse (Figure 2, lanes 2–5), no band appeared in 0.5% cellobiose (Figure 2, lane 1). Therefore, proteomic analysis by LC-MS/MS was performed on 120 kDa, 80 kDa, and 60 kDa bands, respectively.

Comparative proteomics revealed 546 unique proteins across all conditions, with untreated sorghum inducing the broadest response (400 proteins, 73.3%), followed by butanol-pretreated (313 proteins, 57.3%) and NaOH-pretreated (270 proteins, 49.5%) substrates—representing 21.8% and 32.5% reductions in protein diversity, respectively (Figure 3). *K*-means clustering (k = 4) of median-centered log_2_ (PSM + 1) values identified four distinct substrate response patterns: (1) core proteins with stable abundance across substrates (168 proteins, 30.8%), (2) untreated-enriched proteins depleted after pretreatment, (3) NaOH-enriched proteins, and (4) proteins with intermediate responses. Pairwise comparisons confirmed a global shift toward untreated sorghum enrichment, with 57% and 60% of proteins showing higher abundance on untreated versus butanol- and NaOH-pretreated sorghum, respectively. Notably, 169 proteins (31.0%) were exclusively induced by untreated sorghum (Figure 3), representing potential supplementation targets for pretreated biomass saccharification.

The distribution of protein identifications across treatment conditions revealed a clear gradient correlating with substrate complexity and pretreatment severity. Untreated sorghum cultures demonstrated the highest protein diversity, suggesting broad metabolic activation in response to the complex, unmodified lignocellulosic substrate structure. This pattern aligns with the substrate-induced gene regulation model demonstrated in *C. cellulovorans*, where the organism optimizes cellulosomal and non-cellulosomal protein production according to substrate type [1,14]. From the comprehensive proteomic dataset, a total of 54 carbohydrate-active and cellulosomal proteins were selected for detailed analysis, comprising 27 cellulosomal proteins and 27 noncellulosomal proteins (Table 2). The cellulosomal proteins included well-characterized enzymes, such as exoglucanase ExgS (GH48-dockerin sequence), ManA (dockerin-GH5), and EngK (CBM4-GH9-dockerin), as well as the major scaffolding protein CbpA. These cellulosomal enzymes are encoded within a large gene cluster (~22 kb) organized as *cbpA-exgS-engH-engK-hbpA-engL-manA-engM-engN*, previously characterized in *C. cellulovorans* [6,15]. Notably, differential detection of CbpA scaffoldin variants was observed between treatment conditions. The complete CbpA (P38058) containing all nine cohesin domains was detected only in treated sorghum bagasse samples, while a truncated CbpA variant (D9SS73) lacking the ninth C-terminal cohesin was detected across all conditions. The *C. cellulovorans* CbpA scaffoldin contains nine type I cohesin domains for enzyme recruitment, four hydrophilic domains (HLDs), and a family III cellulose-binding domain (CBD) for substrate attachment. The CBD binds crystalline cellulose and chitin with a *Kd* of approximately 1 μM [16]. This differential detection of CbpA variants suggests pretreatment-dependent cellulosome remodeling, a novel observation that extends the substrate-adaptive cellulosome composition reported for this organism [7]. Among cellulosomal proteins, only BglC (GH5-dockerin; D9SW41) was detected exclusively from alkaline-treated sorghum bagasse. In contrast, non-cellulosomal proteins displayed greater condition-specific diversity, including well-characterized enzymes EngO (CBM4-GH9) and EngD (GH5-CBM2) detected across multiple conditions (Table 2). Only two non-cellulosomal enzymes—Bman2A LacZ-CBM-like (A0A173MZW5) and GH127 β-L-arabinofuranosidase (D9STN1)—were detected exclusively from treated sorghum bagasse samples.

### 2.3. Proteomic Analysis of Sorghum-Related Proteins and Their Comparison Based on Untreated Sorghum Bagasse and Its Supernatants, and Sorghum Juice

Anaerobic batch cultivations of *C. cellulovorans* were carried out in a 100 mL medium containing 1% glucose, 0.5% cellobiose, 0.5% sucrose, 1% untreated sorghum bagasse, and 1, 2, 5, and 10% supernatants from untreated sorghum bagasse without shaking. SDS-PAGE analysis revealed similar band patterns in 1% glucose, 0.5% cellobiose, and 1% untreated sorghum bagasse, respectively (Figure 3). Namely, Avicel-bound fractions in each substrate had three major bands, such as 180 kDa, 100 kDa, and 70 kDa, respectively (Figure 4, lanes 2, 5, and 8). On the other hand, Avicel-unbound 120 kDa bands were detected among all substrates (Figure 4, lanes 3, 6, and 9). These results indicated soluble sugars such as glucose and cellobiose might be included and/or generated from untreated sorghum bagasse. Therefore, the supernatants from untreated sorghum bagasse were washed with distilled water and extracted, and then 1, 2, 5, and 10% supernatants in *C. cellulovorans* (C.c) medium were prepared. Surprisingly, the three major bands in Avicel-bound fractions were detected in both 1% and 2% supernatants, whereas no such bands appeared in both 5% and 10% ones (Figure 5). These results suggested that untreated sorghum bagasse contained soluble sugars such as glucose, cellobiose, or sucrose from sorghum juice. On the other hand, whereas 120 kDa bands were detected from all supernatant fractions, high concentration of the supernatants from sorghum bagasse inhibited induction of three major bands in Avicel-bound fractions (Figure 5, lanes 8 and 11). Eventually, 180 kDa, 100 kDa, and 70 kDa bands in Avicel-bound fractions (Figure 4, lane 8) and a 120 kDa band (Figure 3, lane 9) from untreated sorghum bagasse were cut and applied onto LC-MS/MS. Based on identified proteins from untreated sorghum bagasse and its supernatant, Avicel bound or non-bound fractions was shown in Table 3. Four cellulosomal proteins (EngK, Man26A, AidA-GH1-CBM65-DS, and CbpA (D9SS73) were found in both untreated sorghum bagasse and its supernatant. On the other hand, noncellulosomal proteins, only GH43 endo-alpha-1,5-L-arabinanase (D9SQB8) and CBM48-GH13 (D9SVJ6) were detected in the sorghum supernatant. These results suggested that Avicel-bound or -unbound proteins were more limited than the proteins listed in Table 2. More interestingly, five cellulosomal proteins, i.e., Ukcg1-DS, Xyn8A (GH8-DS), PL11-DS, and CBM27-CBM35-like-esterase-DS were not detected from Avicel-bound fractions (180 kDa, 100 kDa, and 70 kDa bands) or an unbound fraction (120 kDa band), suggesting that these cellulosomal enzymes might not be bound to Avicel by themselves. In contrast, noncellulosomal proteins, GH43 endo-alpha-1,5-L-arabinanase and CBM48-GH13 were specifically induced in the supernatant from untreated sorghum bagasse. Accordingly, sorghum juice was used and added into C.c medium containing 1, 2, 3, and 5% (*v*/*v*) sorghum juice. As a result, SDS-PAGE analysis indicated all fractions seemed to have similar patterns of protein bands (Figure 6). Furthermore, the protein expression of Avicel bound bands was dose-dependent on sorghum juice. The bands at 180 kDa, 100 kDa, and 70 kDa in Avicel-bound fractions (Figure 6, lanes 8 and 11) from 3% and 5% sorghum juice were cut and applied onto LC-MS/MS. Between 3% and 5% sorghum juice, the identified cellulosomal proteins seemed similar except for Type-II cohesin (D9SUN3) (Table 3). On the other hand, according to noncellulosomal proteins, alpha-galactosidase GH36 (D9SQL7), arabinofrunosidase GH51 (Q8GEE5), GH43 xylanase (D9SQU9), Lam16B (A0A173MZQ8), and XynB (A0A173N053) were found in 3% sorghum juice, whereas Bgl3A (A0A173MZS9) and Bman2A (A0A173MZW5) were performed in 5% sorghum juice.

### 2.4. Proteomic Analysis of Soluble Sugar-Related Proteins and Their Comparison Based on Untreated Sorghum Bagasse and Its Supernatants, and Sorghum Juice

To compare the identified proteins from untreated or treated sorghum bagasse and its supernatant, three major bands (180 kDa, 100 kDa, and 70 kDa) from soluble sugars (1% (*w*/*v*) glucose, 0.5% (*w*/*v*) cellobiose, and 0.5% (*w*/*v*) sucrose) were investigated by LC-MS/MS. The identified proteins are shown in Table 4. Cellulosomal proteins such as Xyn8A (A0A173MZR7), Eng5C (A0A173N017), ManA (D9SS67), and CBM27-CBM35-like-esterase-DS (D9SWK5) from untreated sorghum bagasse only, rhamnogalacturonan lyase-DS in Avicel-bound protein only from untreated sorghum bagasse, and BglC (D9SW41) from alkaline-treated sorghum bagasse or sorghum juice were not detected in soluble sugars, while these enzymes were only induced by untreated sorghum. On the other hand, nocellulosomal proteins such as alpha-galactosidase GH36 (D9SQL7), CBM4-GH9 (D9SW83), xylose isomerase (D9SR73), and CBM4-GH9 (D9SW83) from treated and untreated sorghum bagasse or sorghum juice, BglB (A0A173MZV), BglD (A0A173MZT0), GH2 LacZ (D9SVV4), and Glycosidase GH130-related protein (D9SUC7) from untreated sorghum bagasse only, Bman2A (A0A173MZW5) from untreated or butanol-treated sorghum bagasse, Arabinofrunosidase GH51 (Q8GEE5) from untreated sorghum bagasse or its supernatant, GH43 xylanase (D9SQU9) from untreated sorghum bagasse or its supernatant or sorghum juice, Lam16B (A0A173MZQ8) from untreated sorghum bagasse or its supernatant, and Bman2A (A0A173MZW5) from untreated sorghum bagasse in Avicel-binding protein or butanol-treated sorghum bagasse, and GH43 endo-alpha-1,5-L-arabinanase (D9SQB8) from sorghum supernatant only were not found in soluble sugars. These results revealed *C. cellulovorans* might sense surrounding sugars or glycans. More interestingly, cellobiose was more limited to induce cellulosomal and noncellulosomal proteins than glucose and sucrose.

## 3. Discussion

In recent decades, the use of lignocellulosic biomass as feedstock for energy production, as well as materials for energy storage, has gained great interest [17]. One of the important reasons for the utilization of biomass relies on its renewability, with a reduction in the net emission of carbon dioxide, and wide distribution with easy availability. Furthermore, cost reductions can be pursued via either in-paradigm or new-paradigm innovation. Sorghum biomass, which serves multiple purposes such as food, forage, and bioenergy feedstock, encounters challenges in maximizing yield, primarily due to a lack of well-characterized biomass-related genes [18]. Two landmark studies have cloned the key gene controlling the juiciness of sweet sorghum stalks [19,20]. In this study, sweet sorghum as a resource biomass has been planted and yielded at the farm field of Mie University. On the other hand, to elucidate the selection of efficient and synergistic enzymes for the degradation and saccharification of cellulosic biomass, molecular networks with neuromorphic architectures may enable molecular decision-making on a level comparable to gene regulatory networks [21,22]. In fact, enzymatic neural networks have so far been developed and brought tangible benefits over non-enzymatic ones, namely speed of operation, compactness of network, composition of computations, sharpness of decision margins, sensitivity of detection, correction of errors, and weighing of analog variables with programmable-gain enzymatic amplification [23]. Recent advances in machine learning have demonstrated potential for predicting enzyme-substrate interactions and optimizing cellulase cocktail compositions for lignocellulose degradation [24,25]. In this context, the proteomic dataset generated in this study, comprising cellulosomal and noncellulosomal proteins from *C. cellulovorans* cultivated on various sorghum-derived substrates, may serve as a foundational resource for future computational approaches aimed at understanding and optimizing sorghum degradation

Consolidated bioprocessing (CBP) between *C. celluloborans* and fermentable microorganisms has been reported and succeeded by using several unused biomasses such as rice strow, sugarcane bagasse, sugar beet pulp [10,26], and mandarin peels and skins [27,28]. In addition, it has been reported that *C. cellulovorans* was able to degrade not only cellulose but also corn fibers and plant cell walls such as cultured tobacco and *Arabidopsis thaliana* by forming their protoplasts [29,30]. In comparison of biomass degradation with rice strow and sugarcane bagasse, the purified cellulosome cultivated from the C.c medium containing rice strow was the best one among the purified cellulosomes prepared from cellobiose, rice strow, and sugarcane bagasse, whereas the purified cellulosome from the C.c medium containing cellobiose was most degradative among them in case of sugarcane bagasse as a substrate [10]. Furthermore, the enzymatic activities of physically treated sugarcane bagasse as a substrate were measured using each enzyme solution from the C.c medium containing pretreated and untreated sugarcane bagasse and cellobiose. As a result, the enzyme solution from the cellobiose medium was the most efficient among the prepared enzyme solutions.

Sorghum bagasse presents a complex lignocellulosic substrate characterized by cellulose (27–45%), hemicellulose (20–27%), and lignin (11–27%) [31,32]. This compositional complexity explains the diverse CAZyme repertoire detected in our proteomics analysis. The higher enzyme diversity in untreated sorghum (400 proteins) compared to pretreated substrates (270–313 proteins) reflects intact structural complexity requiring a broader enzymatic arsenal. Pretreatment removes lignin and solubilizes hemicellulose, reducing substrate heterogeneity and consequently the induced enzyme diversity, providing a mechanistic explanation for the observed 21.8–32.5% reduction in detected proteins. *K*-means clustering confirmed that *C. cellulovorans* mobilizes discrete protein modules in response to substrate complexity, with 57–60% of detected proteins showing higher abundance on untreated sorghum. The 169 untreated-specific proteins represent potential supplementation targets for pretreated biomass saccharification, while the 168-protein core set defines the minimal enzyme complement for sorghum degradation regardless of pretreatment.

In this study, enzyme solutions from *C. cellulovorans* were prepared by treated and untreated sorghum bagasse, its supernatant, and sorghum juice, in addition to soluble sugars such as glucose, cellobiose, and sucrose. At first, it was demonstrated that CMCase activities with enzyme solutions cultivated from cellobiose, filter paper, treated and untreated sorghum bagasse. As a result, the enzyme solution from untreated sorghum bagasse had the highest mean specific activity (6.63 ± 0.82 U/mg protein) rather than other enzyme solutions (Table 1). In contrast, CMCase activity in the enzyme solution of cellobiose culture was 4.70 U/mg, while degradation activity with pretreated sugarcane bagasse from the cellobiose enzyme solution was 0.028 U/mg in the previous study [10]. Thus, enzymatic activity against a target substrate seems dependent on the structure and complexity of cellulose, hemicellulose, and lignin in addition to the content and ratio of carbohydrates. Next, SDS-PAGE analysis was first carried out based on the proteins of culture supernatants, Avicel-bound or unbound fractions. As a result, a variety of identified proteins were observed in untreated sorghum bagasse rather than treated one (Table 2).

The gradient of protein diversity observed in this study (untreated > butanol-treated > alkaline-treated) can be mechanistically interpreted through the lens of substrate-induced gene regulation. *C. cellulovorans* employs substrate recognition systems involving two-component signaling (TCS) and sigma/anti-sigma factor cascades to sense extracellular polysaccharides and regulate enzyme expression accordingly [8]. Native sorghum bagasse—comprising cellulose (27–45%), hemicellulose (20–27%), and lignin (11–27%) [31,33]—presents diverse polysaccharide signals that activate multiple regulatory pathways, resulting in broad enzyme induction (400 proteins). Pretreatment progressively removes these inducing signals: acid–butanol treatment partially solubilizes hemicellulose and disrupts lignin-carbohydrate complexes, reducing xylan- and pectin-derived signals (21.8% protein reduction). Alkaline treatment more extensively removes lignin and solubilizes hemicellulose, leaving primarily cellulose (32.5% protein reduction). This model explains why core cellulases (ExgS, EngK, and EngH) remain constitutively detected as part of the 168-protein core proteome, while accessory hemicellulases and pectinases are progressively lost with increasing pretreatment severity and found among the 169 untreated-specific proteins.

Previous exoproteome analysis of *C. cellulovorans* by Esaka et al. identified 372 proteins in culture supernatants from stationary-phase cultures grown on bagasse, corn germ, or rice straw, selecting 37 cellulosomal and 40 non-cellulosomal proteins for detailed analysis [1]. That study identified biomass-specific proteins, including four bagasse-specific proteins: cellulosomal HbpA (Clocel_2820), and three non-cellulosomal proteins—pectate lyase PL9 (Clocel_0873), α-xylosidase GH31 (Clocel_1430), and glycosidase GH130 (Clocel_3197). Our study extends these findings by examining the effect of industrially relevant pretreatment conditions on the *C. cellulovorans* secretome—an experimental design not previously investigated. While Esaka et al. compared different biomass types (all in native/untreated form) [1], we systematically compared proteome responses to untreated versus chemically pretreated sorghum bagasse, providing mechanistic insight into how substrate accessibility affects enzyme induction.

In particular, Bands 3.1 and 3.3 in SDS-PAGE analysis contained more cellulosomal xylanolytic enzymes such as XynA than observed in treated sorghum bagasse samples (Figure 1). XynA is a key cellulosomal subunit for xylan degradation, possessing both xylanase and acetyl xylan esterase activities that enable efficient attack on acetylated xylans prevalent in grass cell walls [32]. These results indicate that untreated and acid–butanol-treated sorghum bagasse retain substantial xylan content. Effective xylan hydrolysis requires the concerted action of endoxylanases and β-xylosidases [34]. In the case of *C. thermocellum*, integration of a dual-function GH43 xylan hydrolase from *C. clariflavum* enhanced xylan hydrolysis efficiency on corn stalk substrate [35].

In our study, HbpA was detected in Band 3.3 (Table 2) and in the 70 kDa Avicel-bound fraction from untreated sorghum bagasse and the 180 kDa band from sorghum supernatant (Table 3). HbpA possesses a surface layer homology (SLH) domain and a type I cohesin domain, enabling it to bind dockerin-containing cellulosomal enzymes to the cell surface while complementing cellulosome activity [36]. Noncellulosomal PL9 pectate lyase was detected in Band 3.1 (Table 2), the 180 kDa band from untreated sorghum (Table 3), and the 100 kDa and 180 kDa bands from glucose medium (Table 4). Pectate lyases catalyze the breakdown of pectin located in the middle lamella and primary cell wall, leading to maceration of plant tissues [37]. GH31 α-xylosidase was found in Band 4.1 from acid–butanol-treated and Band 5.1 from alkaline-treated sorghum bagasse (Table 2), and the 180 kDa band from 0.5% sucrose medium (Table 4). Interestingly, noncellulosomal GH43 xylanase (D9SQU9) was detected only in untreated sorghum bagasse and 3% sorghum juice (Table 3), suggesting that a dual-function GH43 xylanase could enhance xylan hydrolysis efficiency in sorghum bagasse saccharification. Furthermore, glycosidase GH130 was detected exclusively in Band 3.3 from untreated sorghum bagasse (Table 2). This observation confirms and extends the findings of Esaka et al., who identified GH130 (Clocel_3197) as a biomass-specific enzyme induced during growth on natural soft biomass [1]. GH130 family enzymes are mannoside phosphorylases that act on β-mannosidic linkages in mannan-containing polysaccharides [38,39]. In grasses like sorghum, hemicellulose consists predominantly of substituted xylans (arabinoxylan), with glucomannan present as a minor component. Our novel contribution demonstrates that GH130 expression is pretreatment-dependent, detected only in untreated sorghum and absent after both butanol and alkaline pretreatment. Alkaline pretreatment substantially solubilizes hemicellulose in sorghum bagasse [40], thereby decreasing mannan-containing structures that could serve as inducers of GH130 expression. We propose GH130 as a mannan complexity marker: its presence indicates substrates requiring mannan-degrading capacity in enzyme cocktails, while its absence indicates that hemicellulose has been removed or modified by pretreatment.

The consistent co-detection of ExgS (GH48 exoglucanase) and multiple GH9 endoglucanases (EngH and EngK) as core proteome components across all treatment conditions reflects the synergistic endo–exo mechanism essential for efficient crystalline cellulose degradation. GH9 endoglucanases create new chain ends within cellulose microfibrils by cleaving internal β-1,4-glycosidic bonds, while GH48 exoglucanases processively hydrolyze from these exposed chain ends, releasing cellobiose [41]. This synergy has been quantified in *C. cellulovorans*, where EngH-ExgS combinations assembled with mini-CbpA scaffoldin showed 1.5- to 3-fold higher activity on crystalline cellulose compared to individual enzymes [42]. Notably, synergy was observed only when endoglucanases acted first, followed by ExgS—sequential addition of ExgS first showed almost no synergistic effect.

The pretreatment-dependent protein expression patterns observed in this study have direct implications for industrial enzyme cocktail formulation. Cellobiose is a potent inhibitor of both endoglucanases and exoglucanases [43], and the addition of β-glucosidase to cellulosome preparations has been shown to enhance cellulose degradation up to 10-fold by relieving product inhibition [44]. The detection of multiple β-glucosidases (BglB, Bgl3D, and BglD) exclusively in untreated sorghum indicates that *C. cellulovorans* naturally produces complete cocktails for complex substrates, while β-glucosidase supplementation may be essential for pretreated biomass saccharification.

Based on our findings untreated sorghum bagasse inducing the highest CMCase activity provides actionable guidance for enzyme cocktail design through four interconnected mechanisms: (1) substrate complexity drives comprehensive enzyme induction via regulatory adaptation [45]; (2) CMCase activity serves as a standardizable quality metric for comparing enzyme preparations [46]; (3) high endoglucanase activity initiates the endo–exo synergy required for crystalline cellulose degradation [42]; and (4) native biomass cultivation produces naturally optimized enzyme ratios. These findings suggest that industrial enzyme production for sorghum bagasse should employ native biomass as the inducer substrate to capture the full enzymatic potential of the production organism, with the resulting cocktail either used directly or rationally simplified based on pretreatment severity. Thus, proteomic analysis of *C. cellulovorans* secretome responses provides a systematic approach to selecting optimal enzyme cocktails matched to specific pretreatment conditions and target biomass substrates.

## 4. Materials and Methods

### 4.1. Bacterial Strain and Culture Conditions

*Clostridium cellulovorans* 743B (American Type Culture Collection [ATCC] 35296) was cultivated in a modified anaerobic basal medium under strictly anoxic conditions at 37 °C. The basal medium per 1000 mL consisted of 0.3675 g ammonium chloride, 0.9 g sodium chloride, 0.45 g dipotassium hydrogen phosphate, 0.45 g potassium dihydrogen phosphate, 0.1575 g magnesium chloride hexahydrate, 4.0 g yeast extract, 0.12 g calcium chloride dihydrate, 1 mg resazurin, 1.0 g L-cysteine HCl, 5.0 g sodium bicarbonate, and 5.0 g cellobiose. To ensure sufficient micronutrient availability, the medium was supplemented with 100 mL of trace element solution SL-10, consisting of 8.5 mg manganese chloride tetrahydrate, 9.42 mg cobalt chloride hexahydrate, 52 mg disodium ethylenediaminetetraacetic acid (EDTA), 15 mg iron (II) chloride tetrahydrate, 0.7 mg zinc chloride, 1 mg boric acid, 0.17 mg copper (II) chloride dihydrate, 0.24 mg nickel chloride hexahydrate, 0.36 mg sodium molybdate dihydrate, 66 mg iron (II) sulfate heptahydrate, and 1 g *p*-aminobenzoic acid dissolved in 1000 mL of distilled water. The medium was rendered anaerobically by flushing with CO_2_ gas, and the pH was adjusted to 7.0 ± 0.1 using sterile phosphate buffer. Culture medium preparation was carried out as follows: the primary medium components were dissolved in distilled water, and 1% (*w*/*v*) sorghum-derived substrates or 1% (*w*/*v*) filter paper were incorporated as carbon sources prior to sterilization. The medium was sterilized by autoclaving at 121 °C for 15 min. Heat-labile components, including 1% (*w*/*v*) glucose, 0.5% (*w*/*v*) cellobiose, or 0.5% (*w*/*v*) sucrose, and 1%, 2%, 3%, and 5% (*v*/*v*) sorghum juice, sodium bicarbonate, and L-cysteine HCl, were separately prepared, filter-sterilized under a nitrogen atmosphere, and aseptically added to the cooled medium. Final medium assembly and culturing procedures were conducted within an Anaerobic Chamber (Coy Laboratory Products, Inc., Grass Lake, MI, USA) to maintain anaerobic integrity.

### 4.2. Substrate Preparation

Sorghum species, Kumiai-Hachimitsu (*Sorghum bicolor* L.), was purchased by JACCNET Tokyo, Japan. Sorghum was harvested at physiological maturity from the Mie University agricultural research field. Sorghum bagasse was dried to a constant moisture content (<10%) using an EYELA NDO-450ND (Tokyo, Japan) oven and subsequently ground to a fine particle size with an electrical grinder (Iwatani IFM-800, Osaka, Japan). The dried powder material was subjected to two separate chemical pretreatment protocols. According to acid–butanol pretreatment, sorghum grain powder was treated with 25% (*v*/*v*) 1-butanol and 0.5% (*w*/*w*) H_2_SO_4_ at 200 °C for 60 min. This optimized organosolv pretreatment condition has been demonstrated to achieve the highest cellulose content (84.9%) while maintaining low lignin content (15.3%) [47]. By the alkaline pretreatment, sorghum grain powder was treated with 1% (*w*/*v*) NaOH solution at 121 °C for 60 min in an autoclave, followed by neutralization and extensive washing with distilled water until neutral pH. This treatment, as reported in a previous study, achieved 82.7% lignin removal, reducing the lignin content to 10.9% (*w*/*w*) in the pretreated biomass [48]. Next, the supernatants of 1, 2, 5, and 10% (*w*/*v*) sorghum ground powder were extracted with distilled water. Each supernatant was used for the basal medium. Precultures were established in the same basal medium containing 0.5% (*w*/*v*) cellobiose as the carbon source and incubated for 24 h. 3 mL of the precultures were subsequently used to inoculate 100 mL experimental cultures without shaking.

### 4.3. Enzyme Preparation and Concentration

Culture samples were centrifuged at 8000× *g* for 10 min, 4 °C to obtain cell-free supernatants. Extracellular enzymes were concentrated using ammonium sulfate precipitation by gradually bringing supernatants to 80% saturation with constant stirring at 4 °C. After overnight incubation at 4 °C, samples were centrifuged at 12,000× *g* for 20 min at 4 °C, and protein pellets were resuspended in 50 mM sodium phosphate buffer (pH 6.8). The resuspended proteins were dialyzed against the same buffer using dialysis membrane (molecular mass cutoff 12–14 kDa) at 4 °C for 24 h. The buffer was changed three times. Total protein concentrations were determined using the Bio-Rad Protein Assay kit (Hercules, CA, USA) with bovine serum albumin standards.

### 4.4. SDS-PAGE Analysis and Preparation of Crystalline Cellulose-Bound and Non-Bound Fractions

Sample integrity was verified by SDS-PAGE using the ATTO electrophoresis system (Tokyo, Japan) using pre-cast gradient gels (HERT-520L, ATTO Corporation, Tokyo, Japan) under denaturing conditions. Protein mass marker was purchased and used as Precision Plus Protein™ All Blue Prestained Protein Standards (Bio-Rad). Protein concentrations were normalized to 1.0 μg/μL, and 10 μg of total protein was loaded per lane for all SDS-PAGE analyses. After electrophoresis, gels were stained using AE-1340 EzStain Aqua (ATTO), a Coomassie brilliant blue formulation without organic solvents. Gels were immersed in staining solution for 60 min at room temperature with gentle agitation, then destained with ultrapure water under continuous agitation until protein bands were clearly resolved against a transparent background. The dialyzed solutions concentrated by ammonium sulfate were fractionated into cellulose-binding or cellulose-nonbinding fractions. 100 mg of Avicel PH-101 (Merck, Darmstadt, Germany) was added to 1 mL of each dialyzed solution and stored at 4 °C for 30 min. After centrifugation at 10,000× *g* for 15 min at 4 °C, the supernatant, as the Avicel non-bound fraction, was transferred to a new tube. The Avicel-binding protein pellet was resuspended in 50 mM sodium phosphate buffer (pH 6.8) containing 1 M NaCl. After centrifugation at 10,000× *g* for 15 min at 4 °C, the pellet was recovered and resuspended in 50 mM sodium phosphate buffer (pH 6.8) containing 1M NaCl. After centrifugation at 10,000 × g for 15 min at 4 °C, 100mL of distilled water was added to the pellet as the Avicel-binding fraction.

For proteomic analyses, three different culture conditions (untreated sorghum bagasse, acid–butanol treated sorghum bagasse, and alkaline treated sorghum bagasse) were each cultured once in parallel under identical conditions. For each culture condition, four independent SDS-PAGE analyses were performed to confirm the visual reproducibility of band patterns across technical replicates. Protein bands from two different molecular weight regions (120 kDa and 60 to 80 kDa) were excised from each gel and analyzed separately by LC-MS/MS. In total, each of the three culture conditions was measured twice by LC-MS/MS at two different molecular weight positions, with each molecular weight band collected from four replicate gels to ensure technical reproducibility.

### 4.5. Enzyme Assay

The enzymatic activity of extracellular proteins present in culture supernatants was determined using carboxymethylcellulose (CMC) as the substrate. The reaction mixture consisted of 100 μL of enzyme solution and 900 μL of 0.5% (*w*/*v*) CMC prepared in 50 mM sodium phosphate buffer (pH 6.8), yielding a final reaction volume of 1.0 mL with a concentration of 0.45% (*w*/*v*) CMC. The assays were carried out at 50 °C, which represents the optimal temperature for the activity of the *C. cellulovorans* cellulosome. Following 30 min of incubation, the reactions were terminated by the addition of 3,5-dinitrosalicylic acid (DNS) reagent, and the amount of reducing sugars released was quantified using the DNS method. Enzyme activity was expressed following the standards, where one unit (U) of enzyme activity is defined as the amount of enzyme required to release 1 μmol of glucose equivalent per minute under the assay conditions.

### 4.6. LC-MS/MS Analysis and Data Acquisition

Proteomic analysis was performed using an Orbitrap Fusion Tribrid mass spectrometer (Thermo Fisher Scientific, Waltham, MA, USA) coupled to an EASY-nLC 1000 ultra-performance liquid chromatography (UPLC) system (Thermo Fisher Scientific). Sample preparation followed standard protocols, including tryptic digestion and octadecylsilyl (C18) solid-phase extraction, using the Pierce In-Gel Tryptic Digestion Kit and C18 Spin Tips according to the manufacturer’s instructions (Thermo Fisher Scientific).

#### 4.6.1. Chromatographic Conditions

Peptides were loaded in eluent A (0.1% formic acid) onto a trap column (C18, 3 μm particle size, 2 cm length, 75 μm ID, Acclaim PepMap 100; Thermo Fisher Scientific) and separated using a reverse-phase analytical column (nano high-performance liquid chromatography (HPLC)) capillary column, 75 μm × 12.5 cm, 3 μm, ODS; Nikkyo Technos Co., Ltd., Tokyo, Japan) on a Nanospray Flex Ion Source system (Thermo Fisher Scientific). Separation was achieved with a 40 min gradient using eluents A and B (0.1% formic acid in 80% acetonitrile) at a constant flow rate of 300 nL/min. The gradient was programmed as follows: initial elution at 0% B, a linear increase from 0% to 40% B over 20 min, a rapid increase to 95% B over 2 min, and a final hold at 95% B for 18 min.

#### 4.6.2. Mass Spectrometry Parameters

Data were acquired in data-dependent acquisition mode. Full MS scans were acquired at a resolution of 60,000 with an AGC target in standard mode, a maximum injection time of 50 ms, and a scan range of *m*/*z* 375–1500. MS/MS scans employed quadrupole isolation (isolation window 1.6 *m*/*z*), higher-energy collisional dissociation (HCD, normalized collision energy 30) for fragmentation, and ion trap detection (AGC target in standard mode, maximum injection time 35 ms).

### 4.7. Database Searching and Protein Identification

Raw MS data were processed using Proteome Discoverer 3.0 (Thermo Fisher Scientific) with the Sequest HT search algorithm against the UniProt database for *C. cellovorans* (taxonomy ID: 1493), downloaded on 3 September 2025. A maximum of two missed tryptic cleavage sites was permitted. Precursor and fragment mass tolerances were set to 10 ppm and 0.02 Da, respectively. Methionine oxidation (+15.995 Da) and protein N-terminal acetylation (+42.011 Da) were specified as dynamic modifications, while cysteine carbamidomethylation (+57.021 Da) was defined as a static modification. The Fixed Value PSM Validator was applied for peptide-spectrum match validation. The false discovery rate FDR < 1%.

### 4.8. Statistics

For volumetric and specific activity measurements, pairwise comparisons were performed between each treatment and untreated sorghum using time point and replicate-matched data (n = 15 per comparison). Normality of paired differences was assessed using the Shapiro–Wilk test. Comparisons meeting normality assumptions (*p* > 0.05) were analyzed using paired *t*-tests; non-normal distributions (*p* ≤ 0.05) were analyzed using the Wilcoxon signed-rank test. For specific activity, only the Paper vs. Sorghum comparison violated normality (*p* = 0.001); all other comparisons met normality assumptions.

For proteomics data, *K*-means clustering (*k* = 4) was applied to median-centered log_2_ (PSM + 1) values to identify proteins with similar substrate responses. The Lloyd algorithm with Euclidean distance was used, with multiple random initializations (n_init = 10) and a fixed random seed to improve stability. The solution with the lowest within-cluster variance was retained. Cluster assignments were used to interpret substrate-specific abundance patterns. All statistical analyses were performed using Python 3.12 (pandas, NumPy, scikit-learn, matplotlib, seaborn).

## 5. Conclusions

In this study, enzymatic characterization and proteomic analysis were performed in the culture supernatants from *C. cellulovorans* that contained several sorghum-related substrates, such as untreated or treated sorghum bagasse, its supernatants, and sorghum juice, in comparison with glucose, cellobiose, and sucrose as a simple substrate. CMCase from untreated sorghum bagasse had the highest activity among cellobiose, filter paper as a pure cellulose, untreated and treated sorghum bagasse, suggesting that untreated sorghum bagasse contained not only cellulose, but also hemicellulose and pectin. SDS-PAGE analysis revealed that Avicel-bound fractions from all culture supernatants had three major bands, i.e., 180 kDa, 100 kDa, and 70 kDa, respectively. These bands involved cellulosomal proteins such as the largest scaffolding protein CbpA, cellulases (GH5, GH9, and GH48) and b-glucosidase (GH1), mannanases (GH5 and GH26), xylanases (GH8 and GH10), and pectate lyases (PL6, PL11). Moreover, four bagasse-specific proteins previously reported were identified in this study, including one cellulosomal HbpA and three noncellulosomal proteins, such as PL9 pectate lyase, α-xylosidase GH31, and glycosidase GH130. More interestingly, a dual-function GH43 xylanase belonging to a noncellulosomal protein was found in untreated sorghum bagasse and 3% sorghum juice, respectively.

In conclusion, a variety of cellulosomal and noncellulosomal proteins were induced by glucose, while cellobiose induced more limited and smaller cellulosomal proteins. On the other hand, the identified cellulosomal proteins between 3% and 5% sorghum juice seemed similar except for Type-II cohesin. Furthermore, according to noncellulosomal proteins, alpha-galactosidase GH36, arabinofrunosidase GH51, GH43 xylanase, Lam16B, and XynB were found in 3% sorghum juice, whereas Bgl3A and Bman2A were found in 5% sorghum juice. Thus, the enzymatic system from *C. cellulovorans* would elucidate the degradation of sorghum bagasse, suggesting that the best cocktail of the cellulosome and noncellulosomal enzymes might be obtained in the near future.

## Figures and Tables

**Figure 1 ijms-26-11728-f001:**
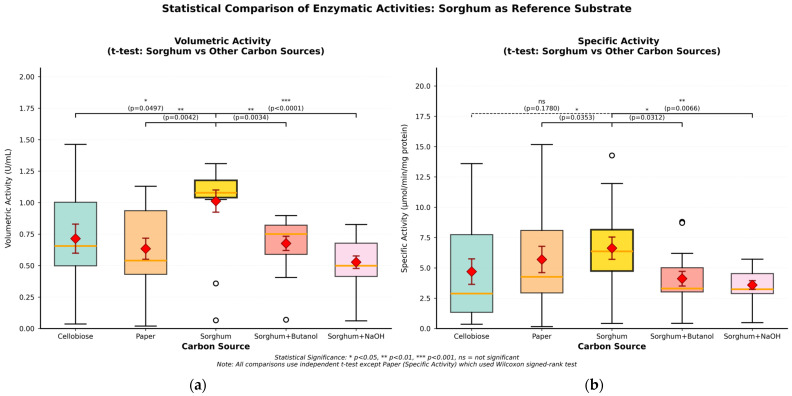
Comparison of CMCase activities cultivated from several carbon sources demonstrates enhanced consistency. (**a**) Volumetric activity distribution (U/mL). (**b**) Specific activity distribution. Box-and-whisker plot showing specific activity (U/mg protein) distributions across all time points (n = 15). Boxes represent interquartile range (25th–75th percentile), horizontal lines indicate median values, diamonds show means ± SEM (error bars), whiskers extend to minimum and maximum values, and open circles denote outliers. The sorghum substrate (highlighted in gold) serves as the reference for all statistical comparisons. Brackets with *p*-values indicate pairwise comparisons between sorghum and other carbon sources using independent samples *t*-tests, except for the Paper-Specific activity comparison, which employed the Wilcoxon signed-rank test due to non-normal distribution. Statistical significance levels are denoted as * *p* < 0.05, ** *p* < 0.01, *** *p* < 0.001, and ns = not significant.

**Figure 2 ijms-26-11728-f002:**
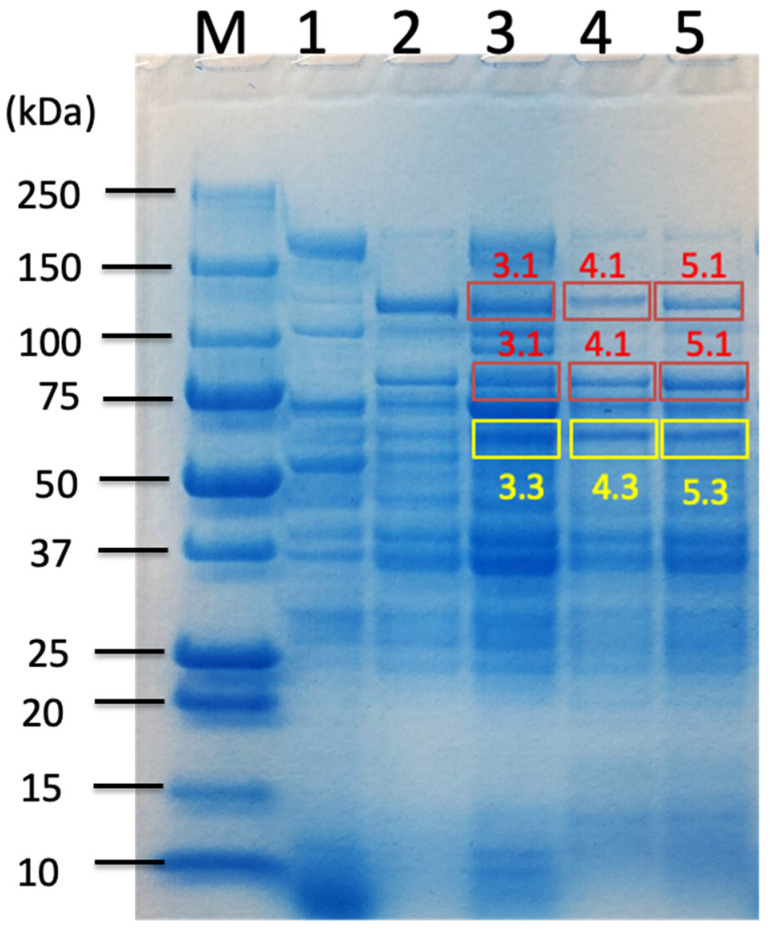
SDS-PAGE of the culture supernatants from *C. cellulovorans* (C.c) medium containing 0.5% cellobiose, 1% filter paper, 1% alkaline-treated sorghum bagasse, and 1% acid–butanol-treated sorghum bagasse. Lane M, molecular mass marker; lane 1, culture supernatant from C.c medium containing 0.5% cellobiose for 2 days; lane 2, culture supernatant from C.c medium containing 1% filter paper for 7 days; lane 3, culture supernatant from C.c medium containing 1% sorghum bagasse for 7 days; lane 4, culture supernatant from C.c medium containing 1% acid–butanol-treated sorghum bagasse for 7 days; lane 5, culture supernatant from C.c medium containing 1% alkaline treated sorghum bagasse for 7 days. Squares of each 3.1, 3.3, 4.1, 4.3, 5.1, and 5.3 indicate cut and fractionated bands, respectively.

**Figure 3 ijms-26-11728-f003:**
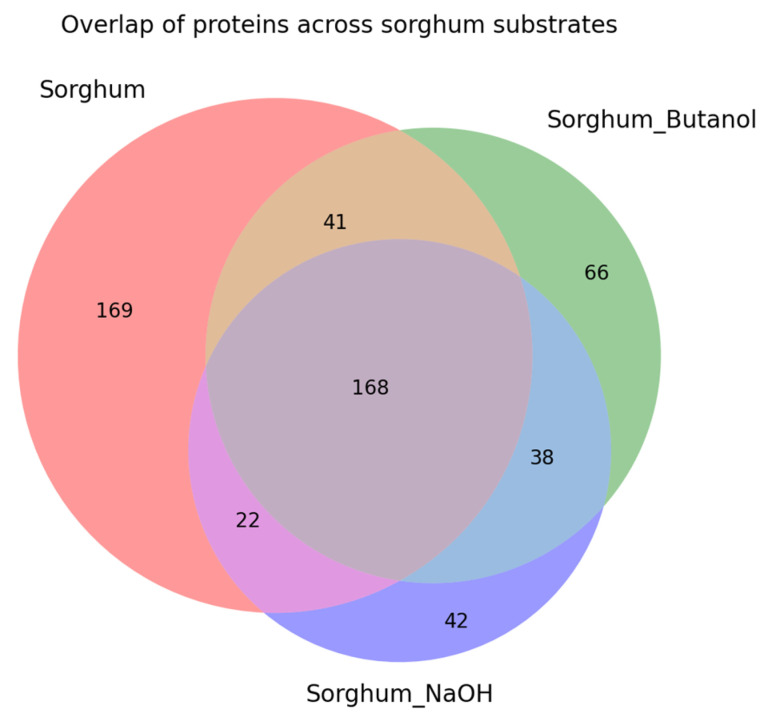
Overlap of *Clostridium cellulovorans* proteins across sorghum substrates. Three-way Venn diagram showing the number of proteins detected on untreated sorghum, butanol-pretreated sorghum, and NaOH-pretreated sorghum. Presence was defined as at least one PSM in a given substrate. Overlapping regions represent proteins shared between two or all three substrates, whereas non-overlapping regions represent proteins unique to a single substrate.

**Figure 4 ijms-26-11728-f004:**
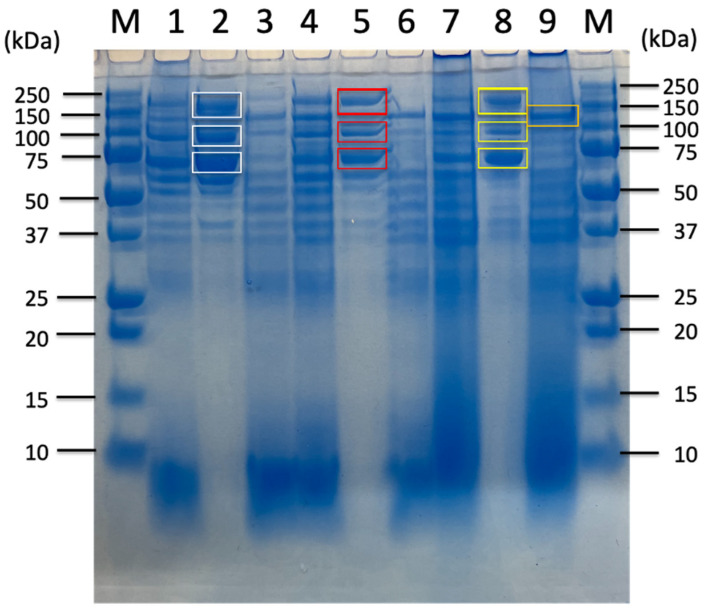
SDS-PAGE of the culture supernatants from *C. cellulovorans* (C.c) medium containing 1.0% glucose, 0.5% cellobiose, and 1% sorghum bagasse. Lanes M, molecular mass marker; lane 1, culture supernatant from C.c medium containing 1% glucose for 2 days; lane 2, Avicel-bound fraction cultured from C.c medium containing 1% glucose for 2 days; lane 3, Avicel non-bound fraction cultured from C.c medium containing 1% glucose for 2 days; lane 4, culture supernatant from C.c medium containing 0.5% cellobiose for 2 days; lane 5, Avicel-bound fraction cultured from C.c medium containing 0.5% cellobiose for 2 days; lane 6, Avicel non-bound fraction cultured from C.c medium containing 0.5% cellobiose for 2 days; lane 7, culture supernatant cultured from C.c medium containing 1% sorghum bagasse for 7 days: lane 8, Avicel-bound fraction cultured from C.c medium containing 1% sorghum bagasse for 7 days; lane 9, Avicel non-bound fraction cultured from C.c medium containing 1% sorghum bagasse for 7 days. Squares of each white, red, and yellow indicate cut and fractionated bands from 180 kDa to 100 kDa to 70 kDa, respectively. The orange square indicates a 120 kDa Avicel non-bound band.

**Figure 5 ijms-26-11728-f005:**
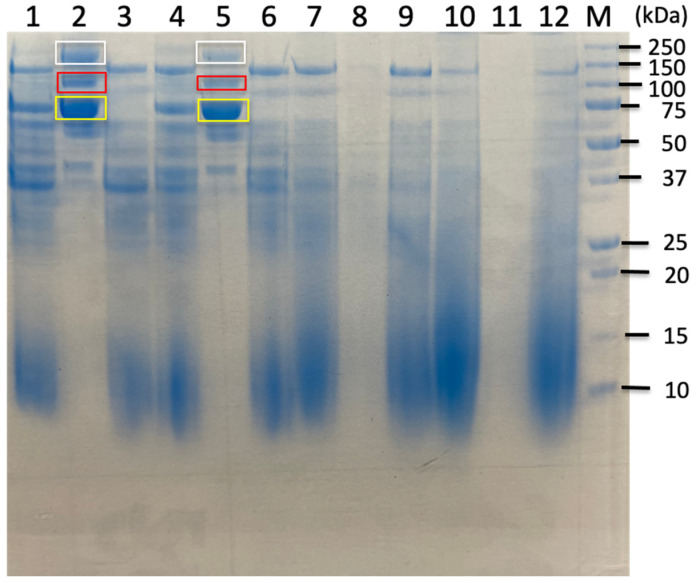
SDS-PAGE of the culture supernatants from *C. cellulovorans* (C.c) medium containing 1, 2, 5, and 10% supernatants extracted from sorghum bagasse. Lane M, molecular mass marker; lane 1, culture supernatant at C.c medium containing 1% supernatant of sorghum bagasse for 4 days; lane 2, Avicel-bound fraction cultured at C.c medium containing 1% supernatant of sorghum bagasse for 4 days; lane 3, Avicel non-bound fraction cultured in C.c medium containing 1% supernatant of sorghum bagasse for 4 days; lane 4, culture supernatant at C.c medium containing 2% supernatant of sorghum bagasse for 7 days; lane 5, Avicel-bound fraction cultured at C.c medium containing 2% supernatant of sorghum bagasse for 7 days; lane 6, Avicel non-bound fraction cultured at C.c medium containing 2% supernatant of sorghum bagasse for 7 days; lane 7, culture supernatant cultured at C.c medium containing 5% supernatant of sorghum bagasse for 7 days: lane 8, Avicel-bound fraction cultured in C.c medium containing 5% supernatant of sorghum bagasse for 7 days; lane 9, Avicel non-bound fraction cultured in C.c medium containing 5% supernatant of sorghum bagasse for 7 days; lane 10, culture supernatant cultured at C.c medium containing 10% supernatant of sorghum bagasse for 7 days: lane 11, Avicel-bound fraction cultured in C.c medium containing 10% supernatant of sorghum bagasse for 7 days; lane 12, Avicel non-bound fraction cultured in C.c medium containing 10% supernatant of sorghum bagasse for 7 days. Squares of each white, red, and yellow indicate cut and fractionated bands from 180 kDa to 100 kDa to 70 kDa, respectively.

**Figure 6 ijms-26-11728-f006:**
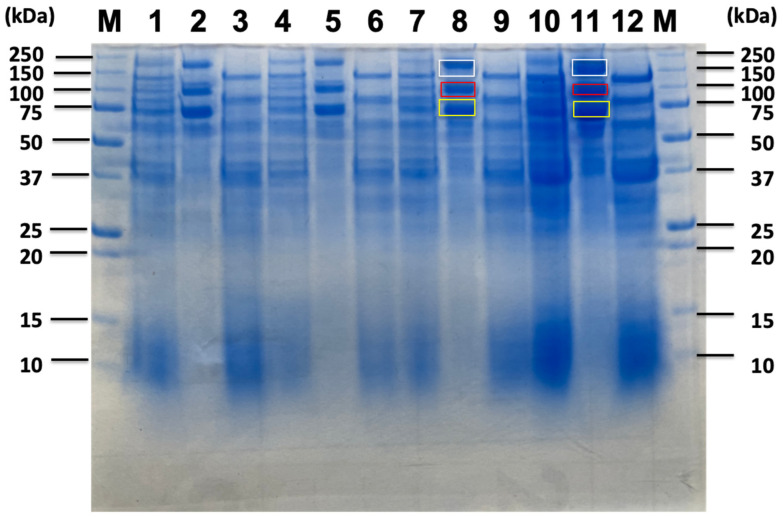
SDS-PAGE of the culture supernatants from *C. cellulovorans* (C.c) medium containing 1, 2, 3, and 5% sorghum juice. Lanes M, molecular mass marker; lane 1, culture supernatant in C.c medium containing 1% sorghum juice for 4 days; lane 2, Avicel-bound fraction cultured in C.c medium containing 1% supernatant of sorghum juice for 4 days; lane 3, Avicel non-bound fraction cultured at C.c medium containing 1% sorghum juice for 4 days; lane 4, culture supernatant at C.c medium containing 2% sorghum juice for 4 days; lane 5, Avicel-bound fraction cultured at C.c medium containing 2% sorghum juice for 4 days; lane 6, Avicel non-bound fraction cultured at C.c medium containing 2% sorghum juice for 4 days; lane 7, culture supernatant cultured at C.c medium containing 3% sorghum juice for 4 days: lane 8, Avicel-bound fraction cultured at C.c medium containing 3% sorghum juice for 4 days; lane 9, Avicel non-bound fraction cultured at C.c medium containing 3% sorghum juice for 4 days; lane 10, culture supernatant cultured at C.c medium containing 5% sorghum juice for 4 days: lane 11, Avicel-bound fraction cultured at C.c medium containing 5% sorghum juice for 4 days; lane 12, Avicel non-bound fraction cultured at C.c medium containing 5% sorghum juice for 4 days. Squares of each white, red, and yellow indicate cut and fractionated bands from 180 kDa, 100 kDa to 70 kDa, respectively.

**Table 1 ijms-26-11728-t001:** CMCase activities from the supernatants cultivated with different carbon sources.

Carbon Sources	Mean Specific Activity (U/mg)	Peak Specific Activity (U/mg)	Time to Peak (h)	Mean Volumetric Activity (U/mL)	Peak Volumetric Activity (U/mL)	Time to Peak (h)
Cellobiose	4.70	13.60	123	0.71	1.46	170
Filter paper	5.70	15.17	146	0.63	1.13	146
Untreated sorghum bagasse	6.63	14.27	123	1.01	1.31	84
Sorghum bagasse + butanol	4.11	8.81	123	0.68	0.90	24
Sorghum bagasse + NaOH	3.59	5.72	84	0.53	0.83	24

**Table 2 ijms-26-11728-t002:** Comparison of the identified cellulosomal and noncellulosomal proteins cultivated from sorghum-related substrates.

		Untreated Sorghum Bagasse		Butanol-Treated Sorghum Bagasse		Alkaline-Treated Sorghum Bagasse	
	*C. cellulovorans* Proteins from Uniplot Database	Band 3.1	Band 3.3	Band 4.1	Band 4.3	Band 5.1	Band 5.3
Cellulosomal proteins	D9SS72_ExgS GH48-DS	●	●	●	●	●	●
	D9SX09_CBM30-GH9-DS	●	●	●	●	●	●
	D9SS71_GH9-CBM3-DS	●	●	●	●	●	●
	A0A173MZP3_Eng9D GH9-DS	●	●	●	●	●	●
	D9SS70_EngK CBM4-GH9-DS	●	●	●	●	●	●
	A0A173N050_Man26A-DS CBM35-GH26-CBM35-CBM35-CBM35-DS	●	●	●	●	●	●
	D9SV64_AidA-GH1-CBM65-DS	●	●	●	●	●	●
	D9SX08_Pectate lyase_PelB-PL6-Helix-DS	●	●	●	●	●	●
	D9SQV8_CBM35-GH26-DS	●	●	●	●	●	●
	D9ST82_GH9-CBM3-DS	●	●	●	●	●	●
	D9SS66_CBM4-GH9-DS	●	●	●	n.d.	●	n.d.
	D9SRK9_GH9-CBM3-DS	●	●	●	n.d.	●	n.d.
	D9SST3_XynA CBM4-GH10-DS	●	●	n.d.	●	n.d.	n.d.
	A0A173N038_Ukcg1-DS	●	●	n.d.	n.d.	n.d.	n.d.
	A0A173MZR7_Xyn8A GH8-DS	●	n.d.	n.d.	n.d.	n.d.	n.d.
	D9STT6_PL11-DS	●	n.d.	n.d.	n.d.	n.d.	n.d.
	D9SWK5_CBM27-CBM35-like-esterase-DS	●	n.d.	n.d.	n.d.	n.d.	n.d.
	D9SS68_GH9-DS	n.d.	●	n.d.	n.d.	n.d.	n.d.
	A0A173N017_Eng5C BglC-DS	n.d.	●	n.d.	n.d.	n.d.	n.d.
	D9SQT1_CBM35-GH26-DS	n.d.	●	n.d.	n.d.	n.d.	n.d.
	D9SS67_ManA DS-GH5	n.d.	●	n.d.	n.d.	n.d.	n.d.
	D9SWK5_CBM27-CBM35-like-esterase-DS	n.d.	●	n.d.	n.d.	n.d.	n.d.
	D9SW41_GH5 BglC-DS	n.d.	n.d.	n.d.	n.d.	●	n.d.
	Schaholdin related:						
	P38058_CbpA	n.d.	n.d.	n.d.	●	●	●
	D9SS73_cohesin-containing protein CbpA	●	●	n.d.	n.d.	n.d.	n.d.
	D9SN69_Cellulosome anchoring protein cohesin region	●	●	●	●	●	●
	D9SR53_Cellulosome anchoring protein cohesin region type-II cohesin	●	●	●	●	●	●
Noncellulosomal proteins	Q6DTY2_EngO CBM4-GH9	●	●	●	●	●	●
	A0A173N0C7_Agal31D	●	●	●	●	●	●
	A0A173MZT5_Axyl31A	●	●	●	●	●	●
	A0A173MZV6_Axyl31B	●	●	●	●	●	●
	D9SR72_Alpha-L-fucosidase GH65	●	●	●	●	●	●
	A0A173MZW6_Agal31A	●	●	●	●	●	n.d.
	D9SQL7_Alpha-galactosidase GH36	●	●	●	●	●	●
	A0A173N0D6_Eng9E GH9	●	●	●	●	●	n.d.
	D9SW83_CBM4-GH9	●	●	●	●	●	●
	D9SR73_Xylose isomerase	●	●	●	●	●	n.d.
	A0A173MZS5_Bgl3C BglX	●	●	●	●	●	●
	A0A173N033_Man26F CBM27-GH26-CBM11	●	●	●	●	●	n.d.
	D9SWQ0_GH5-CBM17	●	●	n.d.	●	n.d.	●
	A0A173N0B1_Bxyl43A	●	●	n.d.	●	n.d.	n.d.
	D9SMP5_GH42 GanA	n.d.	●	●	n.d.	●	●
	D9ST71_GH31	n.d.	n.d.	●	n.d.	●	n.d.
	D9SW84_CBM4-GH9	●	●	n.d.	n.d.	n.d.	n.d.
	A0A173N0E7_Pel1B, PL1, PL9	●	n.d.	n.d.	n.d.	n.d.	n.d.
	A0A173MZV4_BglB	●	●	n.d.	n.d.	n.d.	n.d.
	A0A173MZT4_Bgl3D GH2-BglX	●	n.d.	n.d.	n.d.	n.d.	n.d.
	A0A173MZT0_BglD	●	●	n.d.	n.d.	n.d.	n.d.
	A0A173MZN4_Exo-1,3-beta-glucanase D GH1-CBMX2	●	●	n.d.	n.d.	n.d.	n.d.
	D9SUC7_Glycosidase GH130 related protein (Clocel_3197)	n.d.	●	n.d.	n.d.	n.d.	n.d.
	P28623_EngD GH5-CBM2	n.d.	●	n.d.	n.d.	n.d.	n.d.
	A0A173MZZ7_Epl9B	n.d.	●	n.d.	n.d.	n.d.	n.d.
	A0A173MZW5_Bman2A LacZ-CBM-like	n.d.	n.d.	●	n.d.	n.d.	n.d.
	D9STN1_GH127 beta-L-arabinofuranosidase	n.d.	n.d.	n.d.	●	n.d.	●

● detected; n.d., not detected.

**Table 3 ijms-26-11728-t003:** Comparison of the identified cellulosomal and noncellulosomal proteins cultivated from untreated sorghum and sorghum supernatants.

		Untreated Sorghum Bagasse				Sorghum Supernatnat			3% Sorghum Juice			5% Sorghum Juice		
	Identified Proteins from Sorghum Related Substrates	180 kDa Band (Avicel Bound)	120 kDa Band (Avicel Non-Bound)	100 kDa Band (Avicel Bound)	70 kDa Band (Avicel Bound)	180 kDa Band (Avicel Bound)	100 kDa Band (Avicel Bound)	70 kDa Band (Avicel Bound)	180 kDa Band (Avicel Bound)	100 kDa Band (Avicel Bound)	70 kDa Band (Avicel Bound)	180 kDa Band (Avicel Bound)	100 kDa Band (Avicel Bound)	70 kDa Band (Avicel Bound)
Cellulosomal proteins	D9SS72_ExgS GH48-DS	●	●	●	●	●	n.d.	●	●	●	●	●	●	●
	D9SX09_CBM30-GH9-DS	●	n.d.	●	●	n.d.	●	●	●	●	●	●	●	●
	D9SS71_GH9-CBM3-DS	●	●	●	●	n.d.	●	●	●	●	●	n.d.	●	●
	A0A173MZP3_Eng9D GH9-DS	●	n.d.	●	●	n.d.	●	●	●	●	●	n.d.	●	●
	D9SS70_EngK CBM4-GH9-DS	●	●	●	●	●	●	●	●	●	●	●	●	●
	A0A173N050_Man26A-DS CBM35-GH26-CBM35-CBM35-CBM35-DS	●	●	●	●	●	●	●	●	●	●	●	●	●
	D9SV64_AidA-GH1-CBM65-DS	●	●	●	●	●	●	●	●	●	●	●	●	●
	D9SX08_Pectate lyase_PelB-PL6-Helix-DS	●	●	n.d.	n.d.	●	●	n.d.	●	●	n.d.	●	n.d.	n.d.
	D9SQV8_CBM35-GH26-DS	n.d.	●	n.d.	n.d.	●	●	●	●	●	●	●	●	●
	D9ST82_GH9-CBM3-DS	n.d.	n.d.	n.d.	n.d.	n.d.	●	●	n.d.	n.d.	●	n.d.	●	●
	D9SS66_CBM4-GH9-DS	●	●	●	n.d.	n.d.	●	n.d.	n.d.	●	●	n.d.	●	●
	D9SRK9_GH9-CBM3-DS	●	n.d.	●	●	n.d.	n.d.	●	●	n.d.	●	n.d.	●	●
	D9SST3_XynA CBM4-GH10-DS	n.d.	n.d.	n.d.	●	n.d.	n.d.	●	●	●	●	●	●	●
	A0A173N038_Ukcg1-DS	n.d.	n.d.	n.d.	n.d.	n.d.	n.d.	n.d.	n.d.	n.d.	n.d.	n.d.	n.d.	n.d.
	A0A173MZR7_Xyn8A GH8-DS	n.d.	n.d.	n.d.	n.d.	n.d.	n.d.	n.d.	n.d.	n.d.	n.d.	n.d.	n.d.	n.d.
	D9STT6_PL11-DS	n.d.	n.d.	n.d.	n.d.	n.d.	n.d.	n.d.	n.d.	n.d.	n.d.	n.d.	●	n.d.
	D9SWK5_CBM27-CBM35-like-esterase-DS	n.d.	n.d.	n.d.	n.d.	n.d.	n.d.	n.d.	n.d.	n.d.	n.d.	n.d.	n.d.	n.d.
	D9SS68_GH9-DS	n.d.	n.d.	n.d.	●	n.d.	n.d.	n.d.	●	n.d.	n.d.	n.d.	n.d.	●
	A0A173N017_Eng5C BglC-DS	n.d.	n.d.	n.d.	●	n.d.	n.d.	n.d.	n.d.	n.d.	n.d.	n.d.	n.d.	n.d.
	D9SQT1_CBM35-GH26-DS	n.d.	n.d.	n.d.	●	n.d.	n.d.	n.d.	n.d.	n.d.	n.d.	n.d.	n.d.	n.d.
	D9SS67_ManA DS-GH5	n.d.	n.d.	n.d.	●	●	n.d.	n.d.	●	n.d.	n.d.	n.d.	n.d.	●
	D9SWK5_CBM27-CBM35-like-esterase-DS	n.d.	●	n.d.	●	n.d.	n.d.	n.d.	●	n.d.	n.d.	n.d.	●	n.d.
	D9SW41_GH5 BglC-DS	n.d.	n.d.	n.d.	n.d.	n.d.	●	n.d.	●	n.d.	n.d.	n.d.	n.d.	n.d.
	A0A173MZP8_rhamnogalacturonan lyase-DS	●	n.d.	n.d.	n.d.	n.d.	n.d.	n.d.	n.d.	n.d.	n.d.	n.d.	n.d.	n.d.
	A0A173MZN7_Ukcg2-DS	n.d.	n.d.	●	n.d.	n.d.	n.d.	n.d.	n.d.	n.d.	n.d.	n.d.	n.d.	n.d.
	A0A173N041_Eng5A BglC-DS	n.d.	n.d.	n.d.	n.d.	●	n.d.	n.d.	n.d.	●	●	n.d.	n.d.	●
	D9STQ5 CBMX2-GH5-CBM11-DS	n.d.	n.d.	n.d.	n.d.	●	n.d.	n.d.	●	●	●	n.d.	n.d.	n.d.
	D9SWN8_GH44-DS	n.d.	n.d.	n.d.	n.d.	n.d.	n.d.	●	n.d.	n.d.	●	n.d.	n.d.	●
	D9SVB3_CBM13-CBM35-GH98-DS	n.d.	n.d.	n.d.	n.d.	n.d.	n.d.	n.d.	n.d.	n.d.	n.d.	n.d.	●	n.d.
	A0A173MZX7_Eng5E BglC-DS	n.d.	n.d.	n.d.	n.d.	n.d.	n.d.	n.d.	n.d.	n.d.	n.d.	n.d.	n.d.	●
	Schaholdin related:													
	P38058_CbpA	n.d.	n.d.	n.d.	n.d.	n.d.	n.d.	n.d.	n.d.	n.d.	n.d.	n.d.	n.d.	n.d.
	D9SS73_cohesin-containing protein CbpA	●	●	●	●	●	●	●	●	●	●	●	●	●
	D9SN69_Cellulosome anchoring protein cohesin region	n.d.	n.d.	n.d.	●	n.d.	n.d.	n.d.	n.d.	n.d.	n.d.	n.d.	n.d.	n.d.
	D9SUN3_Type-II cohesin	n.d.	n.d.	n.d.	n.d.	n.d.	n.d.	n.d.	●	●	n.d.	●	n.d.	n.d.
	D9SR53_Cellulosome anchoring protein cohesin region type-II cohesin	n.d.	n.d.	n.d.	n.d.	n.d.	n.d.	n.d.	n.d.	n.d.	n.d.	n.d.	n.d.	n.d.
Noncellulosomal proteins	Q6DTY2_EngO CBM4-GH9	●	n.d.	●	●	●	●	●	●	n.d.	●	n.d.	●	●
	A0A173N0C7_Agal31D	n.d.	n.d.	n.d.	n.d.	n.d.	n.d.	n.d.	n.d.	n.d.	●	n.d.	n.d.	n.d.
	A0A173MZT5_Axyl31A	n.d.	n.d.	n.d.	n.d.	n.d.	n.d.	n.d.	n.d.	n.d.	n.d.	n.d.	n.d.	n.d.
	A0A173MZV6_Axyl31B	n.d.	n.d.	n.d.	n.d.	n.d.	n.d.	n.d.	n.d.	n.d.	n.d.	n.d.	n.d.	n.d.
	D9SR72_Alpha-L-fucosidase GH65	n.d.	n.d.	●	n.d.	n.d.	n.d.	n.d.	n.d.	n.d.	n.d.	n.d.	n.d.	n.d.
	A0A173MZW6_Agal31A	n.d.	n.d.	n.d.	n.d.	n.d.	n.d.	n.d.	n.d.	n.d.	n.d.	n.d.	n.d.	n.d.
	D9SQL7_Alpha-galactosidase GH36	n.d.	n.d.	n.d.	n.d.	n.d.	n.d.	n.d.	n.d.	●	n.d.	n.d.	n.d.	n.d.
	A0A173N0D6_Eng9E GH9	n.d.	●	●	n.d.	n.d.	●	n.d.	n.d.	n.d.	n.d.	n.d.	n.d.	n.d.
	D9SW83_CBM4-GH9	n.d.	●	n.d.	n.d.	n.d.	n.d.	n.d.	n.d.	n.d.	●	n.d.	n.d.	n.d.
	D9SR73_Xylose isomerase	●	●	●	n.d.	n.d.	n.d.	n.d.	n.d.	n.d.	n.d.	n.d.	n.d.	●
	A0A173MZS5_Bgl3C BglX	n.d.	n.d.	n.d.	●	n.d.	n.d.	n.d.	n.d.	n.d.	n.d.	n.d.	n.d.	n.d.
	A0A173N033_Man26F CBM27-GH26-CBM11	●	●	●	n.d.	●	n.d.	n.d.	●	●	●	n.d.	●	n.d.
	D9SWQ0_GH5-CBM17	●	n.d.	●	●	n.d.	●	n.d.	n.d.	n.d.	●	n.d.	n.d.	n.d.
	A0A173N0B1_Bxyl43A	n.d.	●	●	n.d.	n.d.	n.d.	n.d.	n.d.	n.d.	n.d.	n.d.	n.d.	n.d.
	D9SMP5_GH42 GanA	n.d.	n.d.	n.d.	n.d.	n.d.	n.d.	n.d.	n.d.	n.d.	n.d.	n.d.	n.d.	n.d.
	D9ST71_GH31	n.d.	n.d.	n.d.	n.d.	n.d.	n.d.	n.d.	n.d.	n.d.	n.d.	n.d.	n.d.	n.d.
	D9SW84_CBM4-GH9	n.d.	n.d.	n.d.	n.d.	n.d.	n.d.	●	n.d.	n.d.	n.d.	n.d.	n.d.	n.d.
	A0A173N0E7_Pel1B, PL1, PL9	●	n.d.	n.d.	n.d.	n.d.	n.d.	n.d.	n.d.	●	n.d.	●	●	n.d.
	A0A173MZV4_BglB	n.d.	●	n.d.	n.d.	n.d.	n.d.	n.d.	n.d.	n.d.	n.d.	n.d.	n.d.	n.d.
	A0A173MZT4_Bgl3D GH2-BglX	n.d.	●	n.d.	n.d.	n.d.	n.d.	n.d.	n.d.	n.d.	n.d.	n.d.	n.d.	n.d.
	A0A173MZT0_BglD	n.d.	n.d.	n.d.	n.d.	n.d.	n.d.	n.d.	n.d.	n.d.	n.d.	n.d.	n.d.	n.d.
	A0A173MZN4_Exo-1,3-beta-glucanase D GH1-CBMX2	n.d.	n.d.	n.d.	●	n.d.	n.d.	●	●	●	●	●	●	●
	D9SUC7_Glycosidase GH130 related protein (Clocel_3197)	n.d.	n.d.	n.d.	n.d.	n.d.	n.d.	n.d.	n.d.	n.d.	n.d.	n.d.	n.d.	n.d.
	P28623_EngD GH5-CBM2	n.d.	n.d.	n.d.	n.d.	n.d.	n.d.	n.d.	n.d.	n.d.	n.d.	n.d.	n.d.	n.d.
	A0A173MZZ7_Epl9B	n.d.	n.d.	n.d.	n.d.	n.d.	n.d.	n.d.	n.d.	n.d.	n.d.	n.d.	n.d.	n.d.
	A0A173MZW5_Bman2A LacZ-CBM-like	n.d.	n.d.	●	n.d.	n.d.	n.d.	n.d.	n.d.	n.d.	n.d.	n.d.	n.d.	n.d.
	D9STN1_GH127 beta-L-arabinofuranosidase	n.d.	n.d.	n.d.	n.d.	n.d.	n.d.	n.d.	n.d.	n.d.	n.d.	n.d.	n.d.	n.d.
	Q8GEE5_Arabinofrunosidase GH51	●	n.d.	n.d.	n.d.	●	n.d.	n.d.	●	n.d.	n.d.	n.d.	n.d.	n.d.
	D9SQU9_GH43 xylanase	●	n.d.	n.d.	●	n.d.	n.d.	n.d.	●	●	n.d.	n.d.	n.d.	n.d.
	A0A173MZQ8_Lam16B GH16-CBM4-CBM4-CBM4	n.d.	●	●	n.d.	●	●	n.d.	n.d.	n.d.	n.d.	n.d.	n.d.	n.d.
	A0A173N064_Bgal2A	n.d.	●	n.d.	n.d.	n.d.	●	n.d.	n.d.	n.d.	n.d.	n.d.	n.d.	n.d.
	A0A173MZY8_Xyip YcjR	n.d.	●	n.d.	n.d.	n.d.	n.d.	n.d.	n.d.	n.d.	n.d.	n.d.	n.d.	n.d.
	A0A173MZX3_Agal53B GanB	n.d.	●	n.d.	n.d.	n.d.	n.d.	n.d.	n.d.	n.d.	n.d.	n.d.	n.d.	n.d.
	A0A173N093_Agal53A GanB-YdjB-FhaB	n.d.	●	n.d.	n.d.	n.d.	n.d.	n.d.	n.d.	n.d.	n.d.	n.d.	n.d.	n.d.
	A0A173MZS9_Bgl3A BglX-CBM6-like	n.d.	●	●	n.d.	n.d.	●	n.d.	n.d.	n.d.	n.d.	n.d.	●	n.d.
	D9SUV4_AraA L-arabinose isomerase	n.d.	●	n.d.	n.d.	n.d.	n.d.	n.d.	n.d.	n.d.	n.d.	n.d.	n.d.	n.d.
	A0A173MZZ3_Ara43D Afaf	n.d.	●	n.d.	n.d.	n.d.	n.d.	n.d.	n.d.	n.d.	n.d.	n.d.	n.d.	n.d.
	D9SVV4_GH2 LacZ	n.d.	n.d.	●	n.d.	n.d.	n.d.	n.d.	n.d.	n.d.	n.d.	n.d.	n.d.	n.d.
	A0A173MZW5_Bman2A LacZ-CBM-like	n.d.	n.d.	●	n.d.	n.d.	n.d.	n.d.	n.d.	n.d.	n.d.	n.d.	●	n.d.
	D9SQB8_GH43 endo-alpha-1,5-L-arabinanase	n.d.	n.d.	n.d.	n.d.	●	n.d.	n.d.	n.d.	n.d.	n.d.	n.d.	n.d.	n.d.
	D9SVJ6_CBM48-GH13	n.d.	n.d.	n.d.	n.d.	n.d.	●	n.d.	n.d.	n.d.	n.d.	n.d.	n.d.	n.d.
	A0A173N053_XynB Bxyl39B	n.d.	n.d.	n.d.	n.d.	n.d.	n.d.	n.d.	●	n.d.	n.d.	n.d.	n.d.	n.d.

● detected; n.d., not detected.

**Table 4 ijms-26-11728-t004:** Comparison of the identified cellulosomal and noncellulosomal proteins cultivated from soluble sugars.

		1% Glucose Medium			0.5% Celloobiose Medium			0.5% Sucrose Medium		
	Identifiied Proteins from Sorghum Related Substrates	180 kDa Band (Avicel Bound)	100 kDa Band (Avicel Bound)	70 kDa Band (Avicel Bound)	180 kDa Band (Avicel Bound)	100 kDa Band (Avicel Bound)	70 kDa Band (Avicel Bound)	180 kDa Band (Avicel Bound)	100 kDa Band (Avicel Bound)	70 kDa Band (Avicel Bound)
Cellulosomal proteins	D9SS72_ExgS GH48-DS	●	●	●	●	●	●	●	●	●
	D9SX09_CBM30-GH9-DS	●	●	●	n.d.	●	●	n.d.	●	●
	D9SS71_GH9-CBM3-DS	●	●	●	n.d.	n.d.	●	n.d.	●	●
	A0A173MZP3_Eng9D GH9-DS	●	●	●	n.d.	●	●	●	●	●
	D9SS70_EngK CBM4-GH9-DS	●	●	●	●	●	n.d.	●	n.d.	●
	A0A173N050_Man26A-DS CBM35-GH26-CBM35-CBM35-CBM35-DS	●	●	●	●	●	n.d.	●	n.d.	●
	D9SV64_AidA-GH1-CBM65-DS	●	n.d.	●	●	●	●	●	n.d.	●
	D9SX08_Pectate lyase_PelB-PL6-Helix-DS	●	●	n.d.	n.d.	n.d.	n.d.	●	n.d.	n.d.
	D9SQV8_CBM35-GH26-DS	●	●	n.d.	●	●	●	n.d.	●	●
	D9ST82_GH9-CBM3-DS	n.d.	●	●	n.d.	n.d.	●	n.d.	●	●
	D9SS66_CBM4-GH9-DS	●	●	●	n.d.	n.d.	n.d.	n.d.	n.d.	n.d.
	D9SRK9_GH9-CBM3-DS	●	n.d.	●	n.d.	n.d.	●	●	●	●
	D9SST3_XynA CBM4-GH10-DS	●	●	●	n.d.	n.d.	●	n.d.	n.d.	●
	A0A173N038_Ukcg1-DS	n.d.	n.d.	●	n.d.	n.d.	n.d.	n.d.	n.d.	n.d.
	A0A173MZR7_Xyn8A GH8-DS	n.d.	n.d.	n.d.	n.d.	n.d.	n.d.	n.d.	n.d.	n.d.
	D9STT6_PL11-DS	n.d.	n.d.	n.d.	n.d.	●	n.d.	n.d.	n.d.	n.d.
	D9SWK5_CBM27-CBM35-like-esterase-DS	●	●	n.d.	n.d.	●	n.d.	●	●	●
	D9SW41_GH5 BglC-DS	n.d.	n.d.	n.d.	n.d.	n.d.	n.d.	n.d.	n.d.	n.d.
	D9SS68_GH9-DS	n.d.	n.d.	●	n.d.	n.d.	n.d.	n.d.	n.d.	●
	A0A173N017_Eng5C BglC-DS	n.d.	n.d.	n.d.	n.d.	n.d.	n.d.	n.d.	n.d.	n.d.
	D9SQT1_CBM35-GH26-DS	n.d.	n.d.	●	n.d.	n.d.	n.d.	n.d.	n.d.	n.d.
	D9SS67_ManA DS-GH5	n.d.	n.d.	n.d.	n.d.	n.d.	n.d.	n.d.	n.d.	n.d.
	D9SWK5_CBM27-CBM35-like-esterase-DS	n.d.	n.d.	n.d.	n.d.	n.d.	n.d.	n.d.	n.d.	n.d.
	A0A173MZP8_rhamnogalacturonan lyase-DS	n.d.	n.d.	n.d.	n.d.	n.d.	n.d.	n.d.	n.d.	n.d.
	A0A173MZN7_Ukcg2-DS	n.d.	n.d.	●	n.d.	n.d.	n.d.	n.d.	n.d.	n.d.
	D9SWN8_GH44-DS	n.d.	n.d.	●	n.d.	n.d.	n.d.	n.d.	n.d.	●
	A0A173MZX7_Eng5E BglC-DS	n.d.	n.d.	●	n.d.	n.d.	n.d.	n.d.	n.d.	●
	A0A173N041_Eng5A BglC-DS	n.d.	n.d.	●	n.d.	n.d.	n.d.	n.d.	n.d.	n.d.
	D9STQ5_CBMX2-GH5-CBM11-DS	n.d.	n.d.	●	n.d.	n.d.	n.d.	●	n.d.	n.d.
	A0A173MZR1_Non-DS	n.d.	n.d.	n.d.	n.d.	n.d.	n.d.	●	n.d.	n.d.
	D9SVB3_CBM13-CBM35-GH98-DS	n.d.	n.d.	n.d.	n.d.	n.d.	n.d.	●	n.d.	n.d.
	Scaholdin related:									
	P38058_CbpA	n.d.	n.d.	n.d.	n.d.	n.d.	n.d.	n.d.	n.d.	n.d.
	D9SS73_cohesin-containing protein CbpA	●	●	●	●	●	●	●	●	●
	D9SN69_Cellulosome anchoring protein cohesin region	n.d.	n.d.	●	n.d.	n.d.	n.d.	n.d.	n.d.	n.d.
	D9SS69_HbpA Cellulosome anchoring protein cohesin region	n.d.	n.d.	●	n.d.	n.d.	n.d.	n.d.	n.d.	n.d.
	D9SR53_Cellulosome anchoring protein cohesin region type-II cohesin	n.d.	n.d.	n.d.	n.d.	n.d.	n.d.	n.d.	n.d.	n.d.
	D9SUN3 Type-II cohesin	n.d.	n.d.	n.d.	●	●	n.d.	●	n.d.	n.d.
Noncellulosomal proteins	Q6DTY2_EngO CBM4-GH9	n.d.	●	●	n.d.	●	●	n.d.	●	●
	A0A173N0C7_Agal31D_sample3.3	n.d.	n.d.	n.d.	n.d.	n.d.	n.d.	●	n.d.	n.d.
	A0A173MZT5_Axyl31A	n.d.	●	n.d.	n.d.	n.d.	n.d.	n.d.	n.d.	n.d.
	A0A173MZV6_Axyl31B	●	●	n.d.	n.d.	n.d.	n.d.	n.d.	n.d.	n.d.
	D9SR72_GH65 Alpha-L-fucosidase	n.d.	n.d.	n.d.	n.d.	n.d.	n.d.	●	n.d.	●
	A0A173MZW6_Agal31A	n.d.	n.d.	n.d.	n.d.	n.d.	n.d.	●	●	n.d.
	D9SQL7_alpha-galactosidase GH36	n.d.	n.d.	n.d.	n.d.	n.d.	n.d.	n.d.	n.d.	n.d.
	A0A173N0D6_Eng9E GH9	n.d.	●	n.d.	n.d.	●	n.d.	n.d.	n.d.	n.d.
	D9SW83_CBM4-GH9	n.d.	n.d.	n.d.	n.d.	n.d.	n.d.	n.d.	n.d.	n.d.
	D9SR73_Xylose isomerase	n.d.	n.d.	n.d.	n.d.	n.d.	n.d.	n.d.	n.d.	n.d.
	A0A173MZS5_Bgl3C BglX	n.d.	n.d.	n.d.	n.d.	n.d.	n.d.	n.d.	●	n.d.
	A0A173N033_Man26F CBM27-GH26-CBM11	●	●	●	n.d.	●	n.d.	●	n.d.	n.d.
	D9SWQ0_GH5-CBM17	●	●	●	n.d.	●	n.d.	n.d.	n.d.	●
	A0A173N0B1_Bxyl43A	n.d.	n.d.	n.d.	n.d.	●	n.d.	●	n.d.	n.d.
	D9SMP5_GH42 GanA	n.d.	n.d.	●	n.d.	n.d.	n.d.	n.d.	n.d.	n.d.
	D9ST71_GH31	n.d.	n.d.	n.d.	n.d.	n.d.	n.d.	●	n.d.	n.d.
	D9SW84_CBM4-GH9	n.d.	n.d.	n.d.	n.d.	n.d.	n.d.	n.d.	n.d.	n.d.
	A0A173N0E7_Pel1B, PL1, PL9	●	●	n.d.	n.d.	n.d.	n.d.	n.d.	n.d.	n.d.
	A0A173MZV4_BglB	n.d.	n.d.	n.d.	n.d.	n.d.	n.d.	n.d.	n.d.	n.d.
	A0A173MZT4_Bgl3D GH2-BglX	n.d.	n.d.	n.d.	n.d.	n.d.	n.d.	n.d.	n.d.	●
	A0A173MZT0_BglD	n.d.	n.d.	n.d.	n.d.	n.d.	n.d.	n.d.	n.d.	n.d.
	A0A173MZN4_Exo-1,3-beta-glucanase D GH1-CBMX2	n.d.	n.d.	●	n.d.	n.d.	●	n.d.	n.d.	●
	D9SUC7_Glycosidase GH130 related protein (Clocel_3197)	n.d.	n.d.	n.d.	n.d.	n.d.	n.d.	n.d.	n.d.	n.d.
	P28623_EngD GH5-CBM2	n.d.	n.d.	●	n.d.	n.d.	n.d.	n.d.	n.d.	n.d.
	A0A173MZZ7_Epl9B	n.d.	n.d.	●	n.d.	●	n.d.	n.d.	n.d.	n.d.
	A0A173MZZ3_Ara43D Afaf	n.d.	n.d.	n.d.	n.d.	n.d.	n.d.	●	n.d.	n.d.
	D9SUV4_AraA L-arabinose isomerase	n.d.	n.d.	n.d.	n.d.	n.d.	n.d.	n.d.	n.d.	●
	A0A173MZW5_Bman2A LacZ-CBM-like	n.d.	n.d.	n.d.	n.d.	n.d.	n.d.	n.d.	n.d.	n.d.
	D9STN1_GH127 Beta-L-arabinofuranosidase	n.d.	n.d.	n.d.	n.d.	n.d.	n.d.	●	n.d.	n.d.
	Q8GEE5_Arabinofrunosidase GH51	n.d.	n.d.	n.d.	n.d.	n.d.	n.d.	n.d.	n.d.	n.d.
	D9SQU9_GH43 xylanase	n.d.	n.d.	n.d.	n.d.	n.d.	n.d.	n.d.	n.d.	n.d.
	A0A173MZQ8_Lam16B GH16-CBM4-CBM4-CBM4	n.d.	n.d.	n.d.	n.d.	n.d.	n.d.	n.d.	n.d.	n.d.
	A0A173N064_Bgal2A	n.d.	n.d.	n.d.	n.d.	n.d.	n.d.	●	n.d.	●
	A0A173MZS9_Bgl3A BglX-CBM6-like	n.d.	n.d.	n.d.	n.d.	n.d.	n.d.	n.d.	n.d.	●
	D9SVV4_GH2 LacZ	n.d.	n.d.	n.d.	n.d.	n.d.	n.d.	n.d.	n.d.	n.d.
	A0A173MZW5_Bman2A LacZ-CBM-like	n.d.	n.d.	n.d.	n.d.	n.d.	n.d.	n.d.	n.d.	n.d.
	D9SW83_CBM4-GH9	n.d.	n.d.	●	n.d.	●	●	n.d.	n.d.	n.d.
	D9SQB8 GH43 endo-alpha-1,5-L-arabinanase	n.d.	n.d.	n.d.	n.d.	n.d.	n.d.	n.d.	n.d.	n.d.
	D9SVJ6 CBM48-GH13	n.d.	n.d.	n.d.	n.d.	n.d.	n.d.	●	n.d.	n.d.
	D9SW84_CBM4-GH9	n.d.	n.d.	●	n.d.	n.d.	●	n.d.	n.d.	n.d.
	A0A173N053_XynB Bxyl39B	n.d.	n.d.	n.d.	n.d.	n.d.	n.d.	●	n.d.	n.d.
	Q65CL4_PelB	●	n.d.	n.d.	n.d.	n.d.	n.d.	n.d.	n.d.	n.d.
	D9STR9_PL10 Pectate lyase	●	n.d.	n.d.	n.d.	n.d.	n.d.	n.d.	n.d.	●
	A0A173N082_Ara43D GH43-Sugar-binding site	n.d.	n.d.	●	n.d.	n.d.	n.d.	n.d.	n.d.	n.d.
	A0A173N090_AraA	n.d.	n.d.	●	n.d.	n.d.	n.d.	n.d.	●	n.d.
	A0A173N012_EplC	n.d.	n.d.	n.d.	n.d.	●	n.d.	n.d.	n.d.	n.d.
	A0A173N096_Bgal42A GenA	n.d.	n.d.	n.d.	n.d.	●	n.d.	n.d.	n.d.	n.d.
	D9SLW4_GH57 1,4-alpha-glucan branching enzyme	n.d.	n.d.	n.d.	n.d.	n.d.	●	n.d.	n.d.	n.d.
	A0A173MZX6_Amy13C	n.d.	n.d.	n.d.	n.d.	n.d.	●	n.d.	n.d.	n.d.
	A0A173N071_Ukcg4	n.d.	n.d.	n.d.	n.d.	n.d.	●	n.d.	n.d.	n.d.
	A0A173MZW0_Agal31C GH36	n.d.	n.d.	n.d.	n.d.	n.d.	n.d.	●	n.d.	n.d.
	A0A173MZT8_Man26E ManB2	n.d.	n.d.	n.d.	n.d.	n.d.	n.d.	●	n.d.	n.d.
	D9SUU1_GH43 Alpha-L-arabinofuranosidase	n.d.	n.d.	n.d.	n.d.	n.d.	n.d.	n.d.	n.d.	●

● detected; n.d., not detected; 
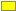
 All substrate; 
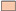
 Untreated soghum only; 
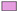
 Untreated sorghum bagasse in Avicel-bound protein only; 
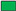
 Butanol-treated or Alkaline-treated sorghum bagasse; 
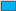
 Alkaline-treated sorghum bagasse or sorghum juice; 
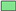
 Sorghum supernatant or sorghum juice; 
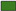
 Untreated sorghum bagasse in Avicel-binding protein or sorghum supernatant, or sorghum juice; 
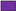
 Sorghum supernatant only; 
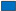
 Sorghum juice only; 
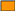
 Soluble sugar only.

## Data Availability

The data supporting the findings of this study are available from the corresponding author upon reasonable request. Due to privacy and ethical restrictions, the data are not publicly available.

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
