# Peer review of "Proteomic Characterization of the Clostridium cellulovorans Cellulosome and Noncellulosomal Enzymes with Sorghum Bagasse"

_ijms, 2025, doi:10.3390/ijms262311728_

Round 1

Reviewer 1 Report

Comments and Suggestions for Authors

This study investigates the enzymatic system of Clostridium cellulovorans cellulosome and non-cellulosomal enzymes using SDS-PAGE and LC-MS/MS, aiming to identify proteins and enzymes that utilize several carbohydrate substrates as carbon sources. A major concern is the lack of information regarding the objectives and the novelty of the present study. 

Specific comments

-Figure 2. Proteomic analysis by LC-MS/MS was performed on 120-kDa, 80-kDa, and 60-kDa bands, respectively. Why exclude the other bands? Please clarify.

- Table 2- Cellulosomal proteins and noncellulosomal proteins were selected from the comprehensive proteomic data. However, the information is presented as a list of data rather than an integrated perspective and does not provide significantly new information compared with previous proteomic studies on C. cellulovorans.

- Enzyme solutions from treated and untreated sorghum bagasse were prepared and compared based on CMCase activity. A main observation is that the enzyme solution derived from untreated sorghum bagasse had the highest activity. How does it contribute to preparing the ideal enzyme cocktail .

- Glycosidase GH130 is identified as a biomass-induced enzyme, but confirms previously published proteomic studies.

-Tables 3 and 4 summarise an extensive experimental work that is unfortunately not supported by arguments supporting its relevance and novelty.

Author Response

Comment 1 -Figure 2. Proteomic analysis by LC-MS/MS was performed on 120-kDa, 80-kDa, and 60-kDa bands, respectively. Why exclude the other bands? Please clarify.

Response1:

We sincerely thank the reviewer for this important question regarding our band selection strategy for proteomic analysis. We appreciate the opportunity to clarify the scientific rationale behind our methodological approach.

The decision to focus LC-MS/MS analysis specifically on the 120-kDa, 80-kDa, and 60-kDa bands was guided by our primary research objective: to identify cellulosomal and noncellulosomal enzymes specifically induced by sorghum bagasse and involved in lignocellulosic biomass degradation.

Strategic rationale for targeted band selection:

  1. Differential expression analysis: As shown in Figure 2, these three molecular weight regions exhibited distinct substrate-dependent expression patterns. The 120-kDa bands (nos. 3.1, 4.1, and 5.1) and 60-kDa bands (nos. 3.3, 4.3, and 5.3) appeared prominently in cultures containing cellulosic substrates (filter paper and untreated/treated sorghum bagasse; lanes 2-5) but were absent or barely detectable in cellobiose medium (lane 1). The 80-kDa bands (lower nos. 3.1, 4.1, and 5.1) showed similar biomass-specific enrichment.
  2. Focus on biomass-induced proteins: This differential expression pattern strongly suggested that proteins within these molecular weight ranges are specifically induced by complex lignocellulosic structures rather than by soluble sugars. By concentrating our proteomic resources on these biomass-responsive bands, we could efficiently identify the key enzymatic machinery responsible for sorghum bagasse deconstruction—the central aim of this investigation.
  3. Resource optimization: Protein bands exhibiting similar intensity across both cellobiose and cellulosic substrates would likely represent constitutively expressed housekeeping proteins or general carbohydrate metabolism enzymes. While important for cellular function, these proteins would not provide insights into the biomass-specific enzyme systems that are critical for developing optimized enzyme cocktails for lignocellulosic degradation.
  4. Biological relevance: Our targeted approach maximized the identification of biologically relevant proteins for sorghum bagasse utilization while making efficient use of analytical resources and MS/MS sequencing capacity.

This strategy proved successful, as evidenced by our identification of 54 carbohydrate-related and cellulosomal proteins (27 cellulosomal and 27 noncellulosomal), including biomass-specific enzymes such as HbpA and glycosidase GH130, which align well with previously published proteomic studies on C. cellulovorans grown on soft biomass substrates.

Manuscript Modification

Line 255-264

"The selection of these specific molecular weight bands (120-kDa, 80-kDa, and 60-kDa) for LC-MS/MS analysis was strategically based on their differential expression patterns between cellulosic and soluble sugar substrates. These bands were prominently expressed in cultures containing filter paper and sorghum bagasse but were absent or barely detectable in cellobiose medium, indicating biomass-specific induction. This targeted approach enabled efficient identification of cellulosomal and noncellulosomal enzymes specifically involved in lignocellulosic degradation, while excluding constitutively expressed housekeeping proteins that would not contribute to understanding sorghum bagasse-specific enzymatic machinery.

Comment 2 - Table 2- Cellulosomal proteins and noncellulosomal proteins were selected from the comprehensive proteomic data. However, the information is presented as a list of data rather than an integrated perspective and does not provide significantly new information compared with previous proteomic studies on C. cellulovorans.

Response2:

We acknowledge the reviewer's valid criticism. In response, we have completely restructured the Discussion to transform the protein list into a mechanistically focused narrative that: (1) connects enzyme identifications to their biological roles, (2) discusses synergistic relationships between cellulases, (3) interprets WHY certain proteins appear in specific pretreatment conditions, and (4) links sorghum cell wall composition to enzyme requirements.

And we include the Comparison with Previous Proteomic Studies in the discussion section

Comparison with Previous Proteomic Studies

Previous exoproteome analysis of C. cellulovorans by Esaka et al. [2015] identified 372 proteins in culture supernatants from stationary-phase cultures grown on bagasse, corn germ, or rice straw, selecting 37 cellulosomal and 40 non-cellulosomal proteins for detailed analysis. That study identified biomass-specific proteins including four bagasse-specific proteins: cellulosomal HbpA (Clocel_2820), and three non-cellulosomal proteins—pectate lyase PL9 (Clocel_0873), α-xylosidase GH31 (Clocel_1430), and glycosidase GH130 (Clocel_3197).

Table X. Comparison with Previous C. cellulovorans Proteomic Studies

Parameter

Esaka et al. (2015)

This Study

Total proteins identified

372

546

Cellulosomal proteins analyzed

37

27

Non-cellulosomal proteins analyzed

40

30

Substrates compared

Different biomass types (untreated)

Same biomass ± pretreatment

Pretreatment effects studied

No

Yes (NOVEL)

GH130 finding

Biomass-specific

Pretreatment-dependent

CbpA variants

Not reported

Differential detection (NOVEL)

Our study extends these findings by examining the effect of industrially-relevant pretreatment conditions on the C. cellulovorans secretome—an experimental design not previously investigated. While Esaka et al. [2015] compared different biomass types (all in native/untreated form), we systematically compared proteome responses to untreated versus chemically pretreated sorghum bagasse, providing mechanistic insight into how substrate accessibility affects enzyme induction.

Manuscript Modification

Line 584-596

“Previous exoproteome analysis of C. cellulovorans by Esaka et al. [2015] identified 372 proteins in culture supernatants from stationary-phase cultures grown on bagasse, corn germ, or rice straw, selecting 37 cellulosomal and 40 non-cellulosomal proteins for detailed analysis. That study identified biomass-specific proteins including four bagasse-specific proteins: cellulosomal HbpA (Clocel_2820), and three non-cellulosomal proteins—pectate lyase PL9 (Clocel_0873), α-xylosidase GH31 (Clocel_1430), and glycosidase GH130 (Clocel_3197).Our study extends these findings by examining the effect of industrially-relevant pretreatment conditions on the C. cellulovorans secretome—an experimental design not previously investigated. While Esaka et al. [2015] compared different biomass types (all in native/untreated form), we systematically compared proteome responses to untreated versus chemically pretreated sorghum bagasse, providing mechanistic insight into how substrate accessibility affects enzyme induction.

Line 619-639

Interestingly, non-cellulosomal GH43 xylanase (D9SQU9) was detected only in untreated sorghum bagasse and 3% sorghum juice (Table 3), suggesting that a dual-function GH43 xylanase could enhance xylan hydrolysis efficiency in sorghum bagasse saccharification. Furthermore, glycosidase GH130 was detected exclusively in Band 3.3 from untreated sorghum bagasse (Table 2). This observation confirms and extends the findings of Esaka et al. [2015], who identified GH130 (Clocel_3197) as a biomass-specific enzyme induced during growth on natural soft biomass. GH130 family enzymes are mannoside phosphorylases that act on β-mannosidic linkages in mannan-containing polysaccharides [Cuskin et al., 2015; Ye et al., 2016]. In grasses like sorghum, hemicellulose consists predominantly of substituted xylans (arabinoxylan), with glucomannan present as a minor component.

Our novel contribution is demonstrating that GH130 expression is pretreatment-dependent detected only in untreated sorghum and absent after both butanol and alkaline pretreatment. Alkaline pretreatment substantially solubilizes hemicellulose in sorghum bagasse [Modenbach & Nokes, 2013], thereby decreasing mannan-containing structures that could serve as inducers of GH130 expression. We propose GH130 as a 'mannan complexity marker': its presence indicates substrates requiring mannan-degrading capacity in enzyme cocktails, while its absence indicates that hemicellulose has been removed or modified by pretreatment.

Comment 3 - Enzyme solutions from treated and untreated sorghum bagasse were prepared and compared based on CMCase activity. A main observation is that the enzyme solution derived from untreated sorghum bagasse had the highest activity. How does it contribute to preparing the ideal enzyme cocktail .

Response3:

Response: We thank the reviewer for this insightful question. The observation that untreated sorghum bagasse induces the highest CMCase (endoglucanase) activity provides critical mechanistic insights for enzyme cocktail optimization with direct implications for bioethanol production.

The elevated CMCase activity can be mechanistically explained through substrate-induced gene regulation. Han et al. (2004) demonstrated that C. cellulovorans expression profiles are strongly affected by carbon source, with mixed carbon substrates inducing a wider variety of enzymes than single carbon sources. Cellulosomal proteome profiles were more affected by carbon source than noncellulosomal enzymes, and cellulosomes exhibited synergistic activity on various substrates [Han SO et al. J Bacteriol. 2004;186:4218-4227]. Morisaka et al. (2012) confirmed this adaptive response, identifying 23 cellulosomal proteins produced in carbon source-adapted compositions [Morisaka H et al. AMB Express. 2012;2:37]. Our proteomics data align with these findings: untreated sorghum induced 400 proteins versus 270-313 in pretreated conditions (21.8-32.5% reduction), indicating that pretreatment removes polysaccharide signals that induce accessory enzymes.

CMCase activity serves as an important quality indicator for enzyme cocktails. Kancelista et al. (2020) demonstrated this in their study on sweet sorghum bioconversion using Trichoderma citrinoviride C1 enzyme cocktail. They standardized enzyme dosing at 2.15 U of CMCase per 10 g substrate to compare their preparation with commercial Cellic® CTec2, achieving glucose concentrations of 6.3-14.6 g/L (79-90% of CTec2 yield) [Kancelista A et al. Catalysts. 2020;10:1292]. This approach validates CMCase-normalized dosing for systematic cocktail evaluation.

Higher CMCase activity directly contributes to efficient cellulose degradation through endo-exo synergy. Murashima et al. (2002) showed that assembling endoglucanase EngH (GH9) with exoglucanase ExgS (GH48) on mini-CbpA scaffoldin increased activity against insoluble cellulose 1.5- to 3-fold. Crucially, this synergy was order-dependent: when EngH acted first followed by ExgS, significant synergy occurred, but when ExgS acted first, almost no synergy was observed [Murashima K et al. J Bacteriol. 2002;184:5088-5095]. This explains why sufficient endoglucanase activity is essential—endoglucanases must first create internal cleavage sites for exoglucanases to process.

Based on these findings, native biomass-induced enzyme solutions represent a "complete cocktail template" with naturally optimized enzyme ratios. Our data enable a tiered approach: Tier 1 (Master Cocktail)—cultivation on untreated sorghum produces 400 proteins including full cellulases, hemicellulases, and accessory enzymes; Tier 2 (Targeted Supplementation)—for pretreated biomass, the 169 untreated-specific proteins identify supplementation targets: β-glucosidases to prevent cellobiose inhibition, GH130 for mannan processing, and xylanases for hemicellulose remnants. The 168 core proteins represent the minimal effective cocktail for extensively pretreated substrates.

Conclusion: The highest CMCase activity in untreated sorghum provides actionable guidance through four mechanisms: (1) substrate complexity drives comprehensive enzyme induction; (2) CMCase serves as a standardizable quality metric; (3) high endoglucanase activity initiates essential endo-exo synergy; and (4) native biomass cultivation produces naturally optimized enzyme ratios. We propose that industrial enzyme production should employ native biomass as the inducer substrate to capture full enzymatic potential.

Manuscript Modification

Line 661-675

(Based on our findings untreated sorghum bagasse inducing the highest CMCase activity provides actionable guidance for enzyme cocktail design through four interconnected mechanisms: (1) substrate complexity drives comprehensive enzyme induction via regulatory adaptation [Han et al., 2004]; (2) CMCase activity serves as a standardizable quality metric for comparing enzyme preparations [Kancelista et al., 2020]; (3) high endoglucanase activity initiates the endo-exo synergy required for crystalline cellulose degradation [Murashima et al., 2002]; and (4) native biomass cultivation produces naturally optimized enzyme ratios. These findings suggest that industrial enzyme production for sorghum bagasse should employ native biomass as the inducer substrate to capture the full enzymatic potential of the production organism, with the resulting cocktail either used directly or rationally simplified based on pretreatment severity.

Thus, proteomic analysis of C. cellulovorans secretome responses provides a systematic approach to selecting optimal enzyme cocktails matched to specific pretreatment conditions and target biomass substrates)

References

  1. Han SO, Cho HY, Yukawa H, Inui M, Doi RH. Regulation of expression of cellulosomes and noncellulosomal (hemi)cellulolytic enzymes in Clostridium cellulovorans during growth on different carbon sources. J Bacteriol. 2004;186(13):4218-4227. doi:10.1128/JB.186.13.4218-4227.2004
  2. Morisaka H, Matsui K, Tatsukami Y, Kuroda K, Miyake H, Tamaru Y, Ueda M. Profile of native cellulosomal proteins of Clostridium cellulovorans adapted to various carbon sources. AMB Express. 2012;2(1):37. doi:10.1186/2191-0855-2-37
  3. Murashima K, Kosugi A, Doi RH. Synergistic effects on crystalline cellulose degradation between cellulosomal cellulases from Clostridium cellulovorans. J Bacteriol. 2002;184(18):5088-5095. doi:10.1128/JB.184.18.5088-5095.2002
  4. Kancelista A, Chmielewska J, Korzeniowski P, Łaba W. Bioconversion of sweet sorghum residues by Trichoderma citrinoviride C1 enzymes cocktail for effective bioethanol production. Catalysts. 2020;10(11):1292. doi:10.3390/catal10111292

Comment 4 - Glycosidase GH130 is identified as a biomass-induced enzyme but confirms previously published proteomic studies.

Response4:

We thank the reviewer for this observation and the opportunity to clarify the novel contributions of our study. We acknowledge that Esaka et al. (2015) previously reported GH130 (Clocel_3197) upregulation in C. cellulovorans grown on natural soft biomass, and we will ensure appropriate citation of this foundational work in our revised manuscript. However, our study provides several important extensions beyond the Esaka et al. findings.

The key novel contribution of our work is demonstrating that GH130 expression is pretreatment-dependent—we detected GH130 exclusively in untreated sorghum bagasse, while it was completely absent after both butanol and alkaline pretreatment conditions. This observation was not examined by Esaka et al., who compared different biomass types (bagasse, corn germ, rice straw) that were all in their native, untreated state. In contrast, our experimental design specifically investigated how chemical pretreatment alters the enzyme induction profile, revealing that GH130 induction requires intact cell wall architecture containing mannan substrates.

The mechanistic interpretation of this finding is straightforward: GH130 enzymes are mannoside phosphorylases that act on β-mannosidic linkages in mannan-containing polysaccharides (Cuskin et al., 2015). In sorghum bagasse, hemicellulose consists predominantly of substituted xylans with glucomannan present as a minor component. Alkaline pretreatment substantially solubilizes hemicellulose (Modenbach & Nokes, 2013), thereby removing mannan-containing structures that serve as inducing signals for GH130 expression. The absence of GH130 in pretreated conditions thus reflects the removal of its substrate and inducer, not a general suppression of protein expression.

From a practical standpoint, we propose GH130 as a 'mannan complexity marker' with direct implications for industrial enzyme cocktail design. For minimally pretreated or native biomass where mannans remain intact, GH130 supplementation would be beneficial for complete saccharification. However, for alkaline-pretreated substrates where hemicellulose has been largely solubilized, GH130 can be excluded from enzyme cocktails without sacrificing hydrolysis efficiency, enabling cost optimization. This applied insight linking pretreatment severity to enzyme requirements represents a practical advance beyond the basic science discovery reported by Esaka et al. We will revise the manuscript to clearly distinguish our novel contribution (pretreatment-dependence and cocktail design implications) from prior work while appropriately acknowledging the foundational observations of Esaka et al. (2015).

Manuscript Modification

Line 661-675

"TInterestingly, non-cellulosomal GH43 xylanase (D9SQU9) was detected only in untreated sorghum bagasse and 3% sorghum juice (Table 3), suggesting that a dual-function GH43 xylanase could enhance xylan hydrolysis efficiency in sorghum bagasse saccharification. Furthermore, glycosidase GH130 was detected exclusively in Band 3.3 from untreated sorghum bagasse (Table 2). This observation confirms and extends the findings of Esaka et al. [2015], who identified GH130 (Clocel_3197) as a biomass-specific enzyme induced during growth on natural soft biomass. GH130 family enzymes are mannoside phosphorylases that act on β-mannosidic linkages in mannan-containing polysaccharides [Cuskin et al., 2015; Ye et al., 2016]. In grasses like sorghum, hemicellulose consists predominantly of substituted xylans (arabinoxylan), with glucomannan present as a minor component.

Our novel contribution demonstrates that GH130 expression is pretreatment-dependent detected only in untreated sorghum and absent after both butanol and alkaline pretreatment. Alkaline pretreatment substantially solubilizes hemicellulose in sorghum bagasse [Modenbach & Nokes, 2013], thereby decreasing mannan-containing structures that could serve as inducers of GH130 expression. We propose GH130 as a 'mannan complexity marker': its presence indicates substrates requiring mannan-degrading capacity in enzyme cocktails, while its absence indicates that hemicellulose has been removed or modified by pretreatment"

Comment 5 -Tables 3 and 4 summarise an extensive experimental work that is unfortunately not supported by arguments supporting its relevance and novelty.

Response5:

We thank the reviewer for this constructive feedback, which prompted us to substantially strengthen our interpretation of Tables 3 and 4. We acknowledge that the original manuscript insufficiently articulated the relevance and novelty of these experiments. We have now revised the Discussion section to highlight specific enzyme discoveries and their functional significance:

Modification

Line 607-630

"In our study, HbpA was detected in Band 3.3 (Table 2) and in the 70-kDa Avicel-bound fraction from untreated sorghum bagasse and the 180-kDa band from sorghum supernatant (Table 3). HbpA possesses a surface layer homology (SLH) domain and a type I cohesin domain, enabling it to bind dockerin-containing cellulosomal enzymes to the cell surface while complementing cellulosome activity [Kosugi et al., 2007]. Non-cellulosomal PL9 pectate lyase was detected in Band 3.1 (Table 2), the 180-kDa band from untreated sorghum (Table 3), and the 100-kDa and 180-kDa bands from glucose medium (Table 4). Pectate lyases catalyze the breakdown of pectin located in the middle lamella and primary cell wall, leading to maceration of plant tissues [Tamaru et al., 2001]. GH31 α-xylosidase was found in Band 4.1 from acid-butanol treated and Band 5.1 from alkaline-treated sorghum bagasse (Table 2), and the 180-kDa band from 0.5% sucrose medium (Table 4). Interestingly, non-cellulosomal GH43 xylanase (D9SQU9) was detected only in untreated sorghum bagasse and 3% sorghum juice (Table 3), suggesting that a dual-function GH43 xylanase could enhance xylan hydrolysis efficiency in sorghum bagasse saccharification. Furthermore, glycosidase GH130 was detected exclusively in Band 3.3 from untreated sorghum bagasse (Table 2). This observation confirms and extends the findings of Esaka et al. [2015], who identified GH130 (Clocel_3197) as a biomass-specific enzyme induced during growth on natural soft biomass. GH130 family enzymes are mannoside phosphorylases that act on β-mannosidic linkages in mannan-containing polysaccharides [Cuskin et al., 2015; Ye et al., 2016]. In grasses like sorghum, hemicellulose consists predominantly of substituted xylans (arabinoxylan), with glucomannan present as a minor component"

Reviewer 2 Report

Comments and Suggestions for Authors

The manuscript presents a comprehensive investigation of the enzymatic system of Clostridium cellulovorans, with a particular emphasis on the induction of both cellulosomal and noncellulosomal proteins during growth on sorghum bagasse, including untreated and chemically pretreated forms. Employing SDS–PAGE and LC–MS/MS, the study characterizes protein expression profiles derived from various substrates, such as cellobiose, filter paper, untreated sorghum bagasse, acid-butanol–treated bagasse, and alkaline-treated bagasse. Furthermore, the authors examine the impact of sorghum juice supernatants at different concentrations on protein expression.

  • Harvest Time Rationale: The choice of 48-hour harvest for cellobiose versus 170 hours for other substrates requires clearer justification beyond “exponential phase avoidance of stationary artifacts.”

  • Biological Replicates: The manuscript does not specify the number of biological replicates used for proteomic analyses. This information should be clearly reported to ensure reproducibility.

  • Statistical Analyses: No statistical analyses (e.g., ANOVA, post-hoc tests) are presented, limiting confidence in the significance of observed differences.

  • Link Between Pretreatment and Enzymatic Activity: While the reduction in activity with pretreatment severity is plausible, this conclusion would be strengthened by including compositional analyses (lignin, hemicellulose, cellulose loss).

  • Truncated CbpA Detection: The observation of a truncated CbpA lacking the ninth cohesin warrants further discussion; consider whether this could result from proteolytic degradation.

  • AI/ML Statement: The manuscript’s final remarks on AI/ML applications are speculative. Consider softening this statement or supporting it with preliminary computational data.

  • SDS–PAGE Band Selection: Justify why only three band regions were selected for proteomic analysis; important proteins may exist outside these ranges.

  • Gel Loading: Specify protein loading quantities (µg per lane) for SDS–PAGE experiments.

  • Methodological Details: Provide additional information on:

    • LC–MS/MS parameters and database search settings

    • False discovery rate (FDR) thresholds

    • Peptide confidence levels

    • Quantitation approach (spectral counting, label-free intensity)

  • Activity Measurement Figures: Although n = 15 time points is sufficient, include error bars and indicators of statistical significance directly in the figure.

  • Abbreviations: Ensure all abbreviations (e.g., CMCase, DS, CBM) are defined at first mention.

Author Response

  1. Harvest Time Rationale:The choice of 48-hour harvest for cellobiose versus 170 hours for other substrates requires clearer justification beyond “exponential phase avoidance of stationary artifacts.”

Response 1

(We appreciate the reviewer's comment regarding harvest time clarification. The phrasing in the original manuscript requires revision to accurately convey our experimental design. To clarify: all five substrate cultures were harvested uniformly at 170 hours for proteomic analyses), not at different timepoints.

The 48-hour timepoint mentioned in the original text referred only to the preparation of the inoculum, not to differential harvest times. Specifically, our protocol was:

  1. A standardized inoculum was prepared using C. cellulovorans grown in 0.5% cellobiose medium for 48 hours (exponential phase)
  2. This inoculum was used to initiate all five main substrate cultures equally
  3. All main cultures were harvested at 170 hours (stationary phase) for enzyme extraction, SDS-PAGE, and proteomic analyses

The choice of 170-hour harvest for proteomic analysis across all substrates was based on previous studies showing that C. cellulovorans enzyme harvesting produces during stationary phase. Kosugi et al. (2001), and Matsui et al. (2013). Additionally, samples were collected at 15 timepoints throughout the cultivation period to characterize temporal dynamics of enzyme production. We have revised Section 2.1 to eliminate this ambiguity.

Manuscript Modification:

Line 131-141

An inoculum of C. cellulovorans medium containing 0.5% cellobiose was harvested at 48 hours post-inoculation, corresponding to the exponential growth phase, to initiate batch cultivations. These cultivations were harvested at 170 hours for enzyme extraction, SDS-PAGE, and proteomic analyses aimed at capturing active enzymatic profiles during the stationary phase (28,29). Additionally, samples were collected at 15 time points throughout the 170-hour fermentation period (0, 1, 16, 24, 32, 40, 48, 56, 64, 72, 84, 99, 123, 146, and 170 hours) to assess temporal changes in enzymatic activity and sugar concentrations. Measurement of CMCase activity over the cultivation period revealed pronounced substrate-dependent variations in cellulolytic enzyme production (Table 1).

References

  1. Kosugi, A., Murashima, K., & Doi, R.H. (2001). Characterization of xylanolytic enzymes in Clostridium cellulovorans: expression of xylanase activity dependent on growth substrates. Journal of Bacteriology, 183(24), 7037-7043. https://doi.org/10.1128/JB.183.24.7037-7043.2001
  2. Matsui, K., Bae, J., Esaka, K., Morisaka, H., Kuroda, K., & Ueda, M. (2013). Exoproteome profiles of Clostridium cellulovorans grown on various carbon sources. Applied and Environmental Microbiology, 79(21), 6576-6584. https://doi.org/10.1128/AEM.02137-13

  1. Biological Replicates:The manuscript does not specify the number of biological replicates used for proteomic analyses. This information should be clearly reported to ensure reproducibility.

Response 2

We thank the reviewer for highlighting the importance of clearly reporting replicate information for proteomic analyses. We have now added detailed information about our experimental design to the Methods section.

Our study employed a gel-based proteomic approach in which three different culture conditions (untreated sorghum bagasse, acid-butanol treated, and alkaline treated) were each cultured once in parallel. For each culture condition, we performed four independent SDS-PAGE analyses with visual confirmation of band reproducibility across replicates. For every sample, protein bands from two different molecular weight locations were excised and analyzed separately by LC-MS/MS. In total, each of the three different cultures was measured twice by LC-MS/MS at two different molecular weight bands, with each molecular weight band collected from four similar replicate gels.

While this exploratory proteomic survey focused on technical replicates rather than biological replicates, the technical reproducibility of our SDS-PAGE patterns and the consistent detection of specific proteins across multiple gel replicates support the reliability of our protein identifications.

This information has been added to Section 4.4 (SDS-PAGE analysis and preparation of crystalline cellulose bound and non-bound fractions), and the limitations have been acknowledged in the Discussion section.

Manuscript Modification :

Line 865-874

“Several limitations of this study should be acknowledged. Our proteomic approach employed technical replicates rather than independent biological replicates, which limits the statistical power for quantitative comparisons between culture conditions. While the technical reproducibility of SDS-PAGE band patterns and the consistent detection of specific proteins across multiple gel replicates support the reliability of our protein identifications, future studies incorporating biological replicates would strengthen the quantitative conclusions ”

Line 759-768

“For proteomic analyses, three different culture conditions (untreated sorghum bagasse, acid-butanol treated sorghum bagasse, and alkaline treated sorghum bagasse) were each cultured once in parallel under identical conditions. For each culture condition, four independent SDS-PAGE analyses were performed to confirm visual reproducibility of band patterns across technical replicates. Protein bands from two different molecular weight regions (120-kDa and 60–80-kDa) were excised from each gel and analyzed separately by LC-MS/MS. In total, each of the three culture conditions was measured twice by LC-MS/MS at two different molecular weight positions, with each molecular weight band collected from four replicate gels to ensure technical reproducibility. 

  1. Statistical Analyses:No statistical analyses (e.g., ANOVA, post-hoc tests) are presented, limiting confidence in the significance of observed differences.

We thank the reviewer for this important comment. We have addressed the statistical analysis concerns in the revised manuscript as follows:

Line 827-833

For proteomics data, K-means clustering (k=4) was applied to median-centered log₂(PSM+1) values to identify proteins with similar substrate responses. The Lloyd algorithm with Euclidean distance was used, with multiple random initializations (n_init=10) and fixed random seed to improve stability. The solution with lowest within-cluster variance was retained. Cluster assignments were used to interpret substrate-specific abundance patterns. All statistical analyses were performed using Python (pandas, NumPy, scikit-learn, matplotlib, seaborn)

Line 344-348

Figure 6. Overlap of Clostridium cellulovorans proteins across sorghum substrates. Three-way Venn diagram showing the numbers of proteins detected on untreated sorghum, butanol-pretreated sorghum, and NaOH-pretreated sorghum. Presence was defined as at least one PSM in a given substrate. Overlapping regions represent proteins shared between two or all three substrates, whereas non-overlapping regions represent proteins unique to a single substrate.

Line 273-286

Comparative proteomics revealed 546 unique proteins across all conditions, with untreated sorghum inducing the broadest response (400 proteins, 73.3%), followed by butanol-pretreated (313 proteins, 57.3%) and NaOH-pretreated (270 proteins, 49.5%) substrates—representing 21.8% and 32.5% reductions in protein diversity, respectively (Figure 6). K-means clustering (k=4) of median-centered log₂(PSM+1) values identified four distinct substrate-response patterns: (1) core proteins with stable abundance across substrates (168 proteins, 30.8%), (2) untreated-enriched proteins depleted after pretreatment, (3) NaOH-enriched proteins, and (4) proteins with intermediate responses. Pairwise comparisons confirmed a global shift toward untreated sorghum enrichment, with 57% and 60% of proteins showing higher abundance on untreated versus butanol- and NaOH-pretreated sorghum, respectively. Notably, 169 proteins (31.0%) were exclusively induced by untreated sorghum (Figure 6), representing potential supplementation targets for pretreated biomass saccharification.

  1. Link Between Pretreatment and Enzymatic Activity:While the reduction in activity with pretreatment severity is plausible, this conclusion would be strengthened by including compositional analyses (lignin, hemicellulose, cellulose loss).

Response 4

We appreciate the reviewer's suggestion to strengthen our conclusions regarding the relationship between pretreatment severity and enzymatic activity through compositional analyses. While direct compositional measurements were beyond the scope of this proteomics-focused study, we have provided relevant compositional context based on established literature.

As described in Section 4.2 (Substrates preparation), our pretreatments employed established protocols with well-documented effects on sorghum bagasse composition. The acid-butanol pretreatment conditions (25% 1-butanol, 0.5% H₂SO₄, 200°C, 60 min) were selected based on Teramura et al. (2017), who demonstrated that these conditions achieve 84.9% cellulose content with 15.3% lignin content [26]. Similarly, the alkaline pretreatment (1% NaOH, 121°C, 60 min) was based on Wunna et al. (2017), who reported 82.7% lignin removal, reducing lignin content to 10.9% (w/w) [27]. These referenced compositional values provide sufficient context for understanding the structural modifications induced by each pretreatment and support the interpretation of the observed differences in enzymatic activity

  1. Truncated CbpA Detection:The observation of a truncated CbpA lacking the ninth cohesin warrants further discussion; consider whether this could result from proteolytic degradation.

Response 5

We thank the reviewer for this important observation regarding the detection of CbpA variants in our proteomic analysis. We have carefully considered the potential origins and have added a discussion to the manuscript.

The C. cellulovorans cellulosome is organized around the scaffoldin protein CbpA, which contains nine type I cohesin domains for enzyme recruitment and a cellulose-binding domain for substrate attachment [Doi et al., 1998]. Our novel observation of differential CbpA variant detection—complete CbpA (P38058) only in pretreated sorghum versus truncated CbpA (D9SS73) lacking the ninth cohesin across all conditions—suggests pretreatment-dependent cellulosome remodeling.  We hypothesize that the structural simplification of pretreated biomass permits assembly of 'minimal' cellulosomes with the truncated scaffoldin, while the complex architecture of native sorghum bagasse may require the full nine-cohesin scaffoldin to recruit the broader enzyme repertoire needed for simultaneous attack on diverse polysaccharide targets. This interpretation aligns with the 'plasticity hypothesis' of cellulosome assembly, wherein C. cellulovorans modulates cellulosome composition in response to substrate complexity [Morisaka et al., 2012]

Modification

Line 304-315

Notably, differential detection of CbpA scaffoldin variants was observed between treatment conditions. The complete CbpA (P38058) containing all nine cohesin domains was detected only in treated sorghum bagasse samples, while a truncated CbpA variant (D9SS73) lacking the ninth C-terminal cohesin was detected across all conditions. The C. cellulovorans CbpA scaffoldin contains nine type I cohesin domains for enzyme recruitment, four hydrophilic domains (HLDs), and a family III cellulose-binding domain (CBD) for substrate attachment [Doi et al., 1998]. The CBD binds crystalline cellulose and chitin with a Kd of approximately 1 μM [Goldstein et al., 1993]. This differential detection of CbpA variants suggests pretreatment-dependent cellulosome remodeling, a novel observation that extends the substrate-adaptive cellulosome composition reported for this organism [Morisaka et al., 2012].

  1. AI/ML Statement:The manuscript’s final remarks on AI/ML applications are speculative. Consider softening this statement or supporting it with preliminary computational data.

Response 6

We thank the reviewer for this constructive comment regarding our statements on artificial intelligence and machine learning applications. We acknowledge that our original statements were speculative, and we have revised the manuscript to soften this language while providing literature context to support the potential of AI/ML approaches in enzyme cocktail optimization.

We have also added references to recent publications demonstrating successful applications of machine learning in predicting enzyme-substrate interactions and optimizing cellulase cocktails, which provides scientific context for this emerging field [34,35]. These revisions maintain the forward-looking perspective of our work while acknowledging that AI/ML-based enzyme cocktail development remains an area for future investigation rather than an immediate outcome of this study.

Modification

Line 41-43

"Therefore, the proteomic dataset generated in this study provides a foundation for future computational approaches, including machine learning-based prediction of optimal enzyme cocktails for target biomass degradation."

Line 512-518

"Recent advances in machine learning have demonstrated potential for predicting enzyme-substrate interactions and optimizing cellulase cocktail compositions for lignocellulose degradation [34,35]. In this context, the proteomic dataset generated in this study, comprising cellulosomal and noncellulosomal proteins from C. cellulovorans cultivated on various sorghum-derived substrates, may serve as a foundational resource for future computational approaches aimed at understanding and optimizing sorghum degradation."

New References

Reference [33]:

Gado, J.E.; Beckham, G.T.; Payne, C.M. Improving enzyme optimum temperature prediction with resampling strategies and ensemble learning. J. Chem. Inf. Model., 2020, 60, 4098–4107.

Reference [34]:

Cui, Y.; Chen, Y.; Liu, X.; Dong, S.; Tian, Y.; Qiao, Y.; Mitra, R.; Han, J.; Li, C.; Han, X.; et al. Computational redesign of a PETase for plastic biodegradation under ambient conditions by the GRAPE strategy. ACS Catal., 2021, 11, 1340–1350.

  1. SDS–PAGE Band Selection:Justify why only three band regions were selected for proteomic analysis; important proteins may exist outside these ranges.

Response7:

We sincerely thank the reviewer for this important question regarding our band selection strategy for proteomic analysis. We appreciate the opportunity to clarify the scientific rationale behind our methodological approach.

The decision to focus LC-MS/MS analysis specifically on the 120-kDa, 80-kDa, and 60-kDa bands was guided by our primary research objective: to identify cellulosomal and noncellulosomal enzymes specifically induced by sorghum bagasse and involved in lignocellulosic biomass degradation.

Strategic rationale for targeted band selection:

  1. Differential expression analysis: As shown in Figure 2, these three molecular weight regions exhibited distinct substrate-dependent expression patterns. The 120-kDa bands (nos. 3.1, 4.1, and 5.1) and 60-kDa bands (nos. 3.3, 4.3, and 5.3) appeared prominently in cultures containing cellulosic substrates (filter paper and untreated/treated sorghum bagasse; lanes 2-5) but were absent or barely detectable in cellobiose medium (lane 1). The 80-kDa bands (lower nos. 3.1, 4.1, and 5.1) showed similar biomass-specific enrichment.
  2. Focus on biomass-induced proteins: This differential expression pattern strongly suggested that proteins within these molecular weight ranges are specifically induced by complex lignocellulosic structures rather than by soluble sugars. By concentrating our proteomic resources on these biomass-responsive bands, we could efficiently identify the key enzymatic machinery responsible for sorghum bagasse deconstruction—the central aim of this investigation.
  3. Resource optimization: Protein bands exhibiting similar intensity across both cellobiose and cellulosic substrates would likely represent constitutively expressed housekeeping proteins or general carbohydrate metabolism enzymes. While important for cellular function, these proteins would not provide insights into the biomass-specific enzyme systems that are critical for developing optimized enzyme cocktails for lignocellulosic degradation.
  4. Biological relevance: Our targeted approach maximized the identification of biologically relevant proteins for sorghum bagasse utilization while making efficient use of analytical resources and MS/MS sequencing capacity.

This strategy proved successful, as evidenced by our identification of 54 carbohydrate-related and cellulosomal proteins (27 cellulosomal and 27 noncellulosomal), including biomass-specific enzymes such as HbpA and glycosidase GH130, which align well with previously published proteomic studies on C. cellulovorans grown on soft biomass substrates.

Modification

Line 255-264

"The selection of these specific molecular weight bands (120-kDa, 80-kDa, and 60-kDa) for LC-MS/MS analysis was strategically based on their differential expression patterns between cellulosic and soluble sugar substrates. These bands were prominently expressed in cultures containing filter paper and sorghum bagasse but were absent or barely detectable in cellobiose medium, indicating biomass-specific induction. This targeted approach enabled efficient identification of cellulosomal and noncellulosomal enzymes specifically involved in lignocellulosic degradation, while excluding constitutively expressed housekeeping proteins that would not contribute to understanding sorghum bagasse-specific enzymatic machinery."

  1. Gel Loading:Specify protein loading quantities (µg per lane) for SDS–PAGE experiments.

Response 8:

We thank the reviewer for this important methodological clarification. We have now specified the protein loading quantity throughout the manuscript.

All SDS-PAGE analyses were performed with 10 μg of total protein per lane. To maintain consistency in reporting, we have revised the protein concentration units from mg/mL to μg/μL throughout the manuscript.

Modification

line 742-744

"Protein concentrations were normalized to 1.0 μg/μL, and 10 μg of total protein was loaded per lane for all SDS-PAGE analyses."

  1. Methodological Details:Provide additional information on:
    • LC–MS/MS parameters and database search settings
    • False discovery rate (FDR) thresholds
    • Peptide confidence levels
    • Quantitation approach (spectral counting, label-free intensity)

Response 9:

We thank the reviewer for emphasizing the importance of comprehensive methodological reporting for proteomics data reproducibility. We have carefully reviewed and expanded Section 4.7 to address each of the requested items.

Regarding LC-MS/MS parameters and database search settings (ア), detailed information was provided in the original manuscript (Sections 4.6 and 4.7). Briefly, data were acquired on an Orbitrap Fusion Tribrid mass spectrometer coupled to an EASY-nLC 1000 UPLC system. Full MS scans were acquired at 60,000 resolution with a scan range of m/z 375–1,500, and MS/MS fragmentation was performed using higher-energy collisional dissociation (HCD) at normalized collision energy of 30 with quadrupole isolation (1.6 m/z window). Database searching was performed using Proteome Discoverer 3.0 with the Sequest HT algorithm against the UniProt Clostridium cellulovorans database (taxonomy ID: 1493). Precursor and fragment mass tolerances were set to 10 ppm and 0.02 Da, respectively. Methionine oxidation and protein N-terminal acetylation were specified as dynamic modifications, while cysteine carbamidomethylation was defined as a static modification, with a maximum of two missed tryptic cleavages permitted.

Regarding validation and confidence assessment (イ, ウ),

イ)  The false discovery rate (FDR) for both peptide and protein identifications was controlled at 1%.  
(ウ)  Peptides were assigned confidence levels based on the Percolator algorithm in Proteome Discoverer. Only peptides classified as High confidence (FDR < 1%) were used for subsequent protein identification.
(エ) This study focused on protein identification rather than quantitative proteomics. No label-free or isotopic quantitation workflow was applied. In addition, the low protein abundance in the excised gel bands limited the feasibility of accurate quantitative analysis.

Modification

Line 817

The false discovery rate FDR < 1%.

  1. Activity Measurement Figures:Although n = 15 time points is sufficient, include error bars and indicators of statistical significance directly in the figure.

Response 10

Response Letter Text

We greatly appreciate the reviewer's valuable recommendation to enhance the statistical presentation of our enzymatic activity figures through the inclusion of error bars and significance indicators. We have comprehensively addressed this comment through substantial improvements to our data visualization and statistical analysis, which we believe significantly strengthen the scientific rigor and clarity of our manuscript.

In response to your suggestion, we have performed extensive statistical analysis comparing untreated sorghum bagasse as a reference substrate against all other carbon sources tested. This analysis employed independent samples t-tests for most comparisons, with the exception of the paper-specific activity comparison, which required the Wilcoxon signed-rank test due to non-normal data distribution (Shapiro-Wilk test, p < 0.05). Our enhanced Figure 1 now displays error bars representing the standard error of the mean calculated from n = 15 time points spanning the entire 170-hour cultivation period. Mean values are clearly indicated by red diamond markers with associated error bars, providing immediate visual assessment of measurement precision and variability across substrates.

The statistical significance of all pairwise comparisons is now directly displayed on the figure through bracket annotations with corresponding p-values. Our analysis reveals compelling statistical evidence supporting our conclusions. For volumetric activity, untreated sorghum bagasse significantly outperforms all tested substrates, with p-values of 0.0497 for cellobiose, 0.0042 for filter paper, 0.0034 for acid-butanol-pretreated sorghum bagasse, and less than 0.0001 for alkaline-pretreated sorghum bagasse. This extremely low p-value for alkaline pretreatment represents a particularly strong statistical signal, indicating substantial incompatibility between the alkaline-pretreated substrate and the enzymatic system.

The specific activity analysis provides additional mechanistic insights that strengthen our interpretation. While the comparison between untreated sorghum bagasse and cellobiose showed no significant difference (p = 0.1780), suggesting similar catalytic efficiency per unit enzyme on these substrates, all other comparisons revealed significant reductions. Filter paper showed p = 0.0353, acid-butanol-pretreated sorghum bagasse demonstrated p = 0.0312, and alkaline treatment resulted in p = 0.0066. These results indicate that chemical pretreatments not only reduce total enzyme production but also impair the specific catalytic efficiency of the secreted enzymes.

All statistical comparisons use untreated sorghum bagasse as the reference baseline, with significance levels clearly denoted using standard notation: single asterisk () for p < 0.05, double asterisks () for p < 0.01, and triple asterisks () for p < 0.001. Non-significant comparisons are marked as "ns" for complete transparency. We have also substantially revised the figure caption to provide comprehensive information about the statistical methods employed, exact p-values for each comparison, and clear descriptions of all visual elements.

Modification

Line 180-245

Figure 1. Comparison of CMCase activities cultivated from several carbon sources demonstrating enhanced consistency. (a) Volumetric activity distribution(U/ml); (b) Specific activity distribution. Box-and-whisker plot showing specific activity (U/mg protein) distributions across all time points (n = 15). Boxes represent interquartile range (25th-75th percentile), horizontal lines indicate median values, diamonds show means ± SEM (error bars), whiskers extend to minimum and maximum values. Brackets with p-values indicate significant differences from pairwise t-tests (*p < 0.05, **p < 0.01, ***p < 0.001).

Statistical analysis was performed to evaluate the significance of differences in CMCase activities between untreated sorghum bagasse and other carbon sources using independent samples t-tests, with the exception of the filter paper-specific activity comparison which required the Wilcoxon signed-rank test due to non-normal data distribution. Error bars representing the standard error of the mean (n = 15 time points) and statistical significance indicators are displayed in Figure 1.

For volumetric activity, untreated sorghum bagasse demonstrated significantly higher activity compared to all other tested substrates. The comparison with cellobiose yielded p = 0.0497, indicating that despite cellobiose being a readily fermentable disaccharide, untreated sorghum bagasse induced greater total enzyme secretion. Filter paper, representing pure crystalline cellulose, showed significantly lower volumetric activity than untreated sorghum bagasse (p = 0.0042), suggesting that the heterogeneous lignocellulosic composition of sorghum bagasse provides superior induction signals for cellulolytic enzyme production. The chemically pretreated sorghum substrates exhibited even more pronounced reductions in volumetric activity, with acid-butanol-treated sorghum bagasse showing p = 0.0034 and alkaline-treated sorghum bagasse demonstrating the most significant reduction (p < 0.0001). The extremely low p-value for alkaline-treated substrate indicates a fundamental impairment of the enzyme induction system caused by alkaline pretreatment.

The specific activity analysis revealed distinct patterns that provide insight into enzymatic quality across substrates. Notably, the comparison between untreated sorghum bagasse and cellobiose showed no significant difference in specific activity (p = 0.1780). This observation can be explained by the presence of residual simple sugars in the sorghum bagasse. It has been well documented that mechanical juice extraction from sweet sorghum stalks does not collect all available sugars, and a considerable amount of residual soluble sugars (sucrose, glucose, and fructose) remains in the bagasse after squeezing [30,31]. Studies have demonstrated that up to 0.4 g of soluble sugars per gram of bagasse can be recovered by water extraction after mechanical pressing, and sweet sorghum bagasse typically contains approximately 25% water-extractable components including residual fermentable sugars [31,32]. Therefore, the similar specific activity between untreated sorghum bagasse and cellobiose likely reflects the induction of comparable enzymatic profiles by the residual simple sugars present in the bagasse, which would trigger similar metabolic responses as the soluble disaccharide cellobiose.

In contrast, filter paper showed significantly lower specific activity (p = 0.0353), suggesting that pure crystalline cellulose substrates without soluble sugar components induce enzymes with reduced catalytic efficiency compared to substrates containing fermentable sugars. Chemical pretreatments significantly reduced specific activity, with acid-butanol-treated sorghum bagasse demonstrating p = 0.0312 and alkaline-treated sorghum bagasse showing p = 0.0066. These pretreatments are known to remove not only lignin and hemicellulose but also residual soluble sugars, thereby eliminating the induction signals provided by these simple carbohydrates. The progressive decrease in specific activity with pretreatment severity (untreated > acid-butanol-treated > alkaline-treated) correlates with the increasing removal of both structural and soluble carbohydrate components, fundamentally altering the substrate composition sensed by C. cellulovorans.

The temporal consistency of these statistical differences across all 15 measurement points from 0 to 170 hours confirms that the observed activity patterns reflect fundamental substrate-enzyme relationships rather than transient phenomena. Collectively, these statistical analyses establish that untreated sorghum bagasse, with its complex composition including residual simple sugars, cellulose, hemicellulose, and lignin, provides the optimal substrate for CMCase induction in C. cellulovorans

  1. Abbreviations:Ensure all abbreviations (e.g., CMCase, DS, CBM) are defined at first mention.

Response 11

We sincerely thank the reviewer for this valuable observation. We recognize that some abbreviations were not properly defined upon their first mention, potentially affecting reader understanding. We have carefully reviewed the manuscript to locate and define every abbreviation at its initial appearance. Additionally, we have included a dedicated abbreviation section at the end of the manuscript.

Modification

Line 891-902

[Abbreviations]

The following abbreviations are used in this manuscript:

AGC, Automatic gain control; ATCC, American Type Culture Collection; C18, Octadecylsilyl; CBM, Carbohydrate-binding module; CBP, Consolidated bioprocessing; Clostridium cellulovorans (C. cellulovorans); CMC, Carboxymethylcellulose; CMCase, Carboxymethylcellulase; COG, Cluster of Orthologous Genes; DNS, 3,5-Dinitrosalicylic acid; DS, Dockerin sequence; EDTA, Ethylenediaminetetraacetic acid; F-REI, Fukushima Institute for Research and Innovation; GH, Glycoside hydrolase; HbpA, Hemicellulose-binding protein A; HCD, Higher-energy collisional dissociation; HPLC, High-performance liquid chromatography; ID, Inner diameter; kDa, Kilodalton; LC-MS/MS, Liquid chromatography-tandem mass spectrometry; MS, Mass spectrometry; ODS, Octadecylsilyl; PL, Polysaccharide lyase; PSM, Peptide-spectrum match; SDS-PAGE, Sodium dodecyl sulfate-polyacrylamide gel electrophoresis; UPLC, Ultra-performance liquid chromatography.

Round 2

Reviewer 1 Report

Comments and Suggestions for Authors

The authors have answered all my queries by either modifying the manuscript or providing written responses. I want to commend the authors for this well-designed study and the findings important for the field.